# Emergent zero-field anomalous Hall effect in a reconstructed rutile antiferromagnetic metal

Meng Wang [1,10] ✉, Katsuhiro Tanaka[2,10], Shiro Sakai[1], Ziqian Wang [1], Ke Deng [3,4], Yingjie Lyu [5], Cong Li[5], Di Tian[5], Shengchun Shen [6], Naoki Ogawa[1,7], Naoya Kanazawa [8], Pu Yu [5], Ryotaro Arita [1,2] & Fumitaka Kagawa[1,9] ✉

The anomalous Hall effect (AHE) that emerges in antiferromagnetic metals shows intriguing physics and offers numerous potential applications. Magnets with a rutile crystal structure have recently received attention as a possible platform for a collinear-antiferromagnetism-induced AHE. $RuO_2$ is a prototypical candidate material, however the AHE is prohibited at zero field by symmetry because of the high-symmetry [001] direction of the Néel vector at the ground state. Here, we show AHE at zero field in Cr-doped rutile, $Ru_{0.8}Cr_{0.2}O_2$. The magnetization, transport and density functional theory calculations indicate that appropriate doping of Cr at Ru sites reconstructs the collinear antiferromagnetism in $RuO_2$, resulting in a rotation of the Néel vector from [001] to [110] while maintaining a collinear antiferromagnetic state. The AHE with vanishing net moment in the $Ru_{0.8}Cr_{0.2}O_2$ exhibits an orientation dependence consistent with the [110]-oriented Hall vector. These results demonstrate that material engineering by doping is a useful approach to manipulate AHE in antiferromagnetic metals.

The Anomalous Hall effect (AHE) long considered as a unique feature of ferromagnetic metals, and its magnitude was empirically taken as proportional to the macroscopic magnetization $M$[1,2]. It followed that in antiferromagnetic materials, which host zero macroscopic magnetization or only small canting moments, the AHE should be negligibly small. However, recent theoretical works indicate that in some antiferromagnetic materials, the AHE can be expected if the magnetic space group (MSG) (or, equivalently, the magnetic point group that the MSG belongs to) allows for a nonzero Berry curvature and/or asymmetric scattering, even if the corresponding macroscopic magnetization is zero[3–5]. Such an AHE was first demonstrated for various noncollinear antiferromagnets with magnetic multipoles[6–12], such as kagome $Mn_3Sn$ and pyrochlore $R_2Ir_2O_7$.

From the symmetry point of view, an antiferromagnetism-induced AHE can also be expected in a collinear antiferromagnet. Recently, such a concept has been proposed in a series of materials[13–15] and experimentally observed in the collinear antiferromagnetic semiconductor $MnTe$[16]. Among these materials, $RuO_2$, which has a rutile structure and exhibits a collinear antiferromagnetic order, has received significant attention as a model

[1]RIKEN Center for Emergent Matter Science (CEMS), Wako 351-0198, Japan. [2]Research Center for Advanced Science and Technology, University of Tokyo, Tokyo 153-8904, Japan. [3]Shenzhen Institute for Quantum Science and Engineering, Southern University of Science and Technology (SUSTech), Shenzhen 518055, China. [4]International Quantum Academy, Shenzhen 518048, China. [5]State Key Laboratory of Low Dimensional Quantum Physics and Department of Physics, Tsinghua University, Beijing 100084, China. [6]Department of Physics, University of Science and Technology of China, Hefei 230026, China. [7]Department of Applied Physics and Quantum-Phase Electronics Center (QPEC), University of Tokyo, Tokyo 113-8656, Japan. [8]Institute of Industrial Science, The University of Tokyo, Tokyo 153-8505, Japan. [9]Department of Physics, Tokyo Institute of Technology, Tokyo 152-8551, Japan. [10]These authors contributed equally: Meng Wang, Katsuhiro Tanaka. ✉e-mail: meng.wang@riken.jp; fumitaka.kagawa@riken.jp

system of the antiferromagnetism-induced AHE[17–20]. As shown in Fig. 1a, the crystal structure of $RuO_2$ consists of two Ru sublattices with antiparallel magnetic moments. The two magnetic sublattices have different chemical environments due to the asymmetric O–Ru–O bond configuration. The simplest argument to determine the presence or absence of the AHE under collinear antiferromagnetism would be to consider how the Hall vector $\sigma_{Hall} = (\sigma_{yz}, \sigma_{zx}, \sigma_{xy})$ is transformed by the symmetry operations[14,18]; here, note that $\sigma_{yz}$, $\sigma_{zx}$, and $\sigma_{xy}$ represent only anti-symmetric part of the conductivity tensor. When the Néel vector ($L$) of $RuO_2$ is along the [110] direction, the MSG is $Cmm'm'$, in which $\sigma_{Hall}$ along [110] is invariant under all symmetry operations and thus allows for a zero-field AHE[14]. In contrast, if $L \parallel$ [001], the MSG is $P4'_2/mnm'$, which does not allow for a finite $\sigma_{Hall}$ because no vector can be invariant under two orthogonal rotation symmetry operations (see Supplementary Note 1 for details)[14]. A previous neutron diffraction indicates that the Néel vector in $RuO_2$ is along [001][17], and hence $\sigma_{Hall}$ and the zero-field AHE are prohibited by symmetry (Supplementary Fig. 1).

To unveil the AHE associated with the collinear antiferromagnetism in $RuO_2$, a recent study focused on tilting the Néel vector from [001] toward [110] by utilizing a high magnetic field of ~50 T[19,20]. This phenomenon can be viewed as a magnetic-field-induced AHE associated with a Néel vector, forming a sharp contrast to AHEs in ferromagnets[1,2], in which the AHE can be observed even under zero field. Achieving a zero-field AHE in such a rutile-type collinear antiferromagnet remains a major challenge for experiments.

Previous density functional theory (DFT) calculations have revealed that the easy axis of the Néel vector in $RuO_2$ sensitively depends on the electron filling[19,20], which inspired us to pursue the zero-field AHE in the derivatives of $RuO_2$ by means of appropriate modulations on its Fermi level. To change the direction of the Néel vector from [001] and therefore render the zero-field AHE allowed by symmetry, we dope Cr into $RuO_2$. Since the $4d$ orbital level of $Ru^{4+}$ is slightly higher than the $3d$ orbital level of $Cr^{4+}$, a charge transfer from $Ru^{4+}$ to $Cr^{4+}$ ions is expected (Fig. 1b)[21,22] while favouring antiparallel spin coupling between the nearest-neighboring Ru and Cr sites. Considering that collinear spin orders are realized in both $RuO_2$ (antiferromagnetic) and $CrO_2$ (ferromagnetic) in rutile phases[17,23,24], the collinear antiferromagnetic state is reasonably expected in stoichiometric proximity to $RuO_2$.

It should be noted that the collinear antiferromagnetism in $RuO_2$ has been questioned quite recently[25]. Nevertheless, considering that many previous experiments support the collinear antiferromagnetism[17,19,24,26,27], we designed the experiment by postulating that the magnetism of $RuO_2$ at the ground state is a collinear antiferromagnetic state with the Néel vector along [001] and interpret the experimental results with the assumption that a small amount of Cr-doping does not change the collinear antiferromagnetism but can modulate the direction of the Néel vector. Within this approach, our magnetometry suggests that the direction of the Néel vector in the $Ru_{0.8}Cr_{0.2}O_2$ film shifts to [110]. Concomitantly, we find that the $Ru_{0.8}Cr_{0.2}O_2$ film exhibits an appreciable

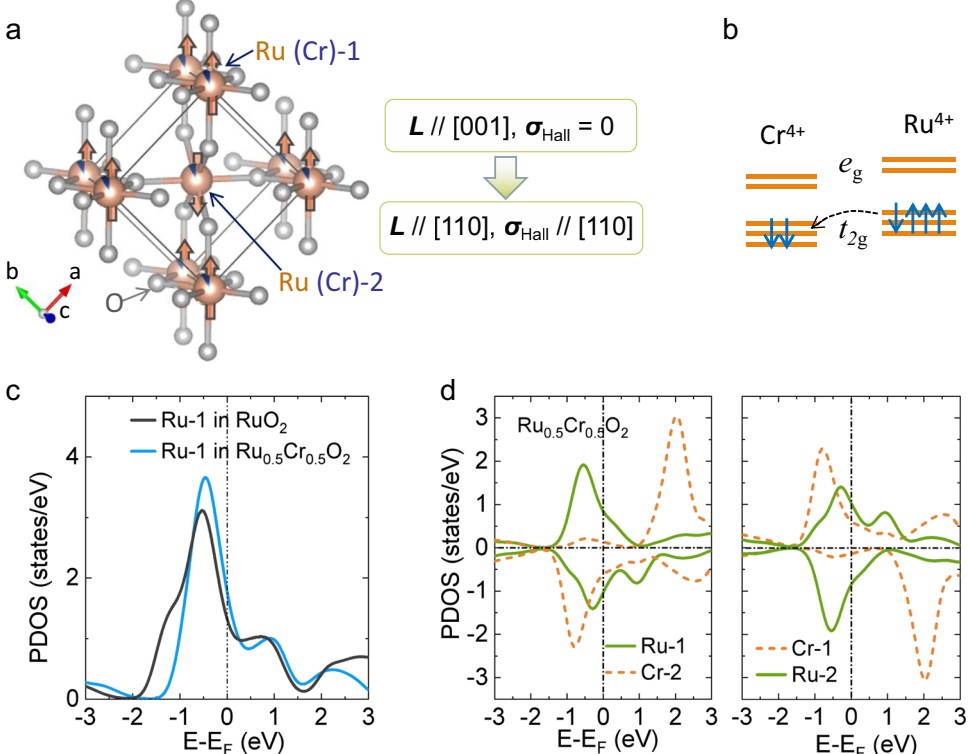

**Fig. 1 | Antiferromagnetic symmetry controlled anomalous Hall effect (AHE) and DFT calculations for Cr-doped RuO₂. a** Crystal structure of the Cr-doped rutile phase $RuO_2$. O-ions are located between two Ru (Cr) sites asymmetrically. The Ru-1 (Cr-1) and Ru-2 (Cr-2) denote the Ru (Cr) ions at the center and the corner sites of the unit cell, respectively. The orange arrows denote the local magnetic moment with antiferromagnetic coupling along [110]. Hall vector ($\sigma_{Hall}$) is allowed and parallel to the Néel vector ($L$) along [110] in such a configuration, which vanishes as the Néel vector is along [001], indicating a manipulating of $L$ is necessary to generate AHE. **b** Schematic illustration of charge transfer in Cr-doped $RuO_2$. The orbital level

difference between the nearest neighbor sites can lead to partial charge transfer from $Ru^{4+}$ to $Cr^{4+}$ to form a reconstructed Fermi level and maintain an antiparallel spin coupling. **c** Calculated projected density of states (PDOS) of the $RuO_2$ and $Ru_{0.5}Cr_{0.5}O_2$ in the paramagnetic phase. The Ru-2 sites for both components possess identical PDOS with Ru-1. **d** Calculated PDOS of the $Ru_{0.5}Cr_{0.5}O_2$ in the magnetic ground state. The doped Cr ions have two selective sites as labeled by Cr-1 and Cr-2 in (**a**). Ru and Cr both show an asymmetric PDOS (a spontaneous polarization), while exhibiting an antiparallel coupling.

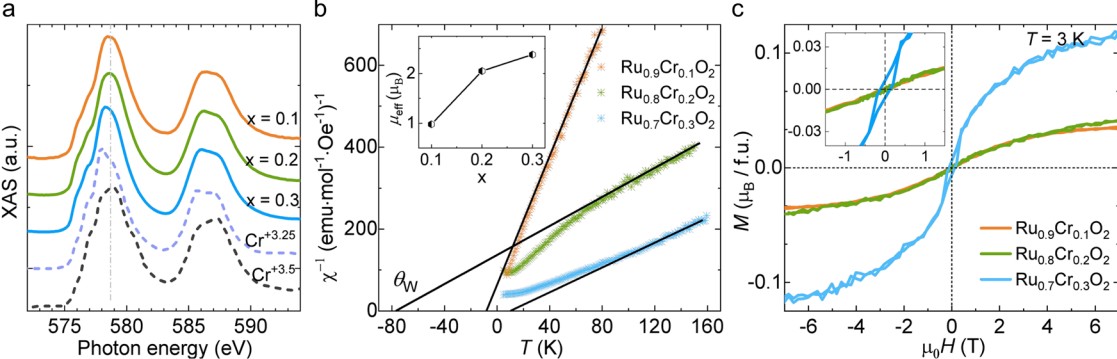

**Fig. 2 | XAS and magnetic states evolution in $Ru_{1-x}Cr_xO_2$ films grown on $TiO_2$ (110). a** XAS around the $L$-edge of Cr measured in the $Ru_{1-x}Cr_xO_2$ films compared to that in $La_{1-x}Sr_xCrO_3$[28]. **b, c** Temperature-dependent magnetic susceptibility (**b**) and magnetic field-dependent magnetization (**c**) curves measured with a magnetic field along the out-of-plane (OOP) axis. All films are grown on $TiO_2$ (110). Inset of (**b**), the effective on-site moments ($\mu_{eff}$) depending on the doping level x.

Inset of (**c**), an expanded view of the low-field region. Linear fittings of the $\chi^{-1}$–$T$ curves at high temperatures indicate an antiferromagnetic behavior with negative Weiss temperatures $\theta_W = -10$ K and $-75$ K in x = 0.1 and 0.2, respectively. The x = 0.3 film shows a small positive $\theta_W$, with a finite remanent magnetization at zero field, implying a ferrimagnetic ground state. The magnetic field is 1 T for the $\chi^{-1}$–$T$ measurement.

zero-field AHE with hysteretic behavior while the net magnetization is vanishingly small. These observations are well explained by considering that the collinear antiferromagnetism with the Néel vector along [110] is realized and that the magnetic field switches the two collinear antiferromagnetic states that are related by time-reversal operation.

## Results

### DFT calculations on the impact of Cr-doping

To gain insight into the impact of Cr-doping on the Fermi level, we first performed DFT calculations for the paramagnetic states of $Ru_{1-x}Cr_xO_2$ for x = 0 and 0.5. As shown in Fig. 1c, by doping Cr, the shift of the projected density of states (or, equivalently, the shift of the Fermi level) is observed, as expected. The magnetic calculation for x = 0.5 (Fig. 1d) further demonstrates that the ground state has appreciable local magnetic moments with antiparallel couplings among the nearest-neighboring Cr and Ru ions. Note that the DFT + $U$ calculations on $RuO_2$ show that the energy difference with the Néel vector orienting to [001], [100], and [110] is tiny (~5 meV) and that the easy-axis direction sensitively depends on the Fermi level (Supplementary Fig. 2)[19]. Our DFT results therefore support our working hypothesis that Cr doping is a promising approach to change the Néel vector direction while maintaining the collinear antiferromagnetic order.

The DFT+DMFT results indicate that Cr doping is also accompanied by the enhancement of the local magnetic moment. For the case of non-doped $RuO_2$, the Ru ions exhibit a negligibly small spin polarization when $U$ is small (<1 eV) (Supplementary Fig. 3). In contrast, when Cr is doped, considerable local moments are observed (0.15 $\mu_B$ for x = 0.25 and 0.4 $\mu_B$ for x = 0.5; see Supplementary Fig. 3) in the DFT calculations, even at $U = 0$.

Thus, based on our DFT calculations, we can expect that the easy axis of the Néel vector changes from the original [001] direction, in which the zero-field AHE is prohibited. These expectations are verified by the experiments described below.

### Films fabrication and valence evaluation

We synthesized the $Ru_{1-x}Cr_xO_2$ films by pulsed laser deposition (PLD) on $TiO_2$ (110) substrates with x = 0.1, 0.2, and 0.3 (see "Methods"). The high crystalline quality of the films was confirmed by X-ray $2\theta$-$\omega$ scans (see supplementary Fig. 4a) and the surface topography with atomic terraces (Supplementary Fig. 4c). Besides, the resistivities of the materials increase as the doping level increases, while all compounds show a metallic behavior, as shown in Supplementary Fig. 5a. The robust metallicity implies the strong overlap of Cr and Ru orbitals.

To probe the valence state of the doped Cr in the rutile lattice, we carried out soft X-ray absorption spectroscopy (XAS) measurements (see "Methods") on the three films. Figure 2a shows the XAS results near the $L$-edge of Cr, with a comparison to that from $La_{1-x}Sr_xCrO_3$ materials[28]. The Cr in all of the $Ru_{1-x}Cr_xO_2$ films exhibits a fractional valence state between +3.25 and +3.5. As the doping level increases from 0.1 to 0.3, the peak shows a gradual shift to lower energy, indicating a gradual decrease in valence. Such a result is consistent with our scenario that the Cr doping is accompanied by the charge transfer and the corresponding Fermi-level shift.

### Antiferromagnetic metal phases in the $Ru_{1-x}Cr_xO_2$ films

To check whether the magnetic ground state is still antiferromagnetic upon the Cr doping, we performed magnetic susceptibility ($\chi$) and magnetization ($M$) measurements with magnetic field ($H$) and temperature ($T$) dependences (see "Methods" and Supplementary Fig. 6 for details). The results are summarized in Figs. 2b, c, and we first focus on the results of x = 0.1 and 0.2. The high-temperature regions of the $\chi^{-1}$–$T$ profiles are fitted with the Curie–Weiss law, $\chi = C/(T-\theta_W)$, and we obtain $\theta_W \approx -10$ K and $-75$ K for x = 0.1 and 0.2, respectively. These results indicate that an antiferromagnetic interaction is dominant in x = 0.1 and 0.2[29–32]. Moreover, the effective on-site moments ($\mu_{eff}$) obtained from the fittings are distinctly enhanced with increasing Cr-doping levels (Fig. 2b, inset and Supplementary Note 2)[29,30], which is also consistent with our DFT calculations.

The $M$–$H$ curves at the lowest temperature, 3 K, demonstrate that the spontaneous net magnetization at zero field is too small to be distinguished in the antiferromagnetic $Ru_{0.9}Cr_{0.1}O_2$ and $Ru_{0.8}Cr_{0.2}O_2$ (Fig. 2c). Moreover, the field-induced moment at 7 T is only 0.03 $\mu_B$ (x = 0.1) and 0.04 $\mu_B$ (x = 0.2) per formula unit ($\mu_B$/f.u.), which are almost two orders of magnitude smaller than that in ferromagnetic $SrRuO_3$ and $CrO_2$[33–35], excluding the possibility of a ferromagnetic ground state for x = 0.1 and 0.2.

In the $Ru_{0.7}Cr_{0.3}O_2$ film, contrastingly, the analysis based on the Curie–Weiss law results in a small positive $\theta_W$ with $\mu_{eff}$ of ~2.5 $\mu_B$ per site (Fig. 2b, and Supplementary Note 2). Furthermore, the $M$–$H$ curve exhibits a finite remanent magnetization, and the magnetization at 7 T is distinctly larger compared with the case of x = 0.1 and 0.2. These observations indicate the evolution of a ferrimagnetic phase in x = 0.3, consistent with the tendency from $RuO_2$ to $CrO_2$[17,23,24,26]. Therefore, the AHE accompanying the ferrimagnetic phase in x = 0.3 is beyond the scope of this study.

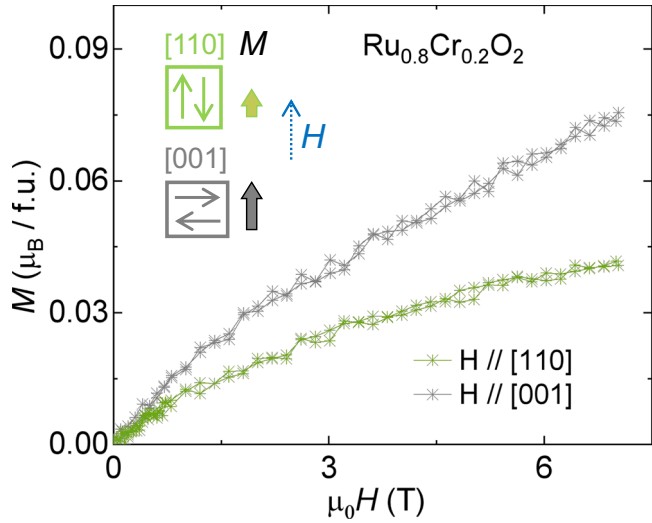

**Fig. 3 | Magnetic anisotropy and Néel vector orientation in Ru$_{0.8}$Cr$_{0.2}$O$_2$ film grown on TiO$_2$ (110).** M–H curves were measured at 3 K with magnetic field along out-of-plane (H ∥ [110]) and in-plane (H ∥ [001]), respectively. Inset, the illustration of spin orientation and corresponding moments (wide arrows) driven by the external magnetic field (H) applied to [110] and [001].

## Néel-vector direction in the Ru$_{0.8}$Cr$_{0.2}$O$_2$ film

We then focus on the antiferromagnetic Ru$_{0.8}$Cr$_{0.2}$O$_2$ (110) sample and aim to reveal the direction of the Néel vector. The DFT calculations in RuO$_2$ suggest a finite net magnetic moment when the Néel vector along [100] is assumed (Supplementary Fig. 2a), which should be preserved in the doped phase. Our M–H measurements in Ru$_{0.8}$Cr$_{0.2}$O$_2$ show a vanishing net moment, thereby ruling out the possibility that the Néel vector is along [100]. Then, the remaining candidates of the Néel-vector direction are the [001] and [110] orientations. To test these two possibilities, we refer to the fact that the field-induced moment in a collinear antiferromagnet is generally minimized when the field is parallel to the Néel vector, as illustrated in Fig. 3 inset[19,30]. The anisotropy of the field-induced moment was measured on the Ru$_{0.8}$Cr$_{0.2}$O$_2$ (110) film for the fields of the out-of-plane [110] and in-plane [001] directions. The anisotropic response demonstrates that the [110] axis exhibits a smaller field-induced moment (Fig. 3), suggesting that the Néel vector is likely along [110], rather than [001], in Ru$_{0.8}$Cr$_{0.2}$O$_2$, although other orientations cannot be ruled out completely. Note that if the Néel vector along [110] is realized, the zero-field AHE is allowed by symmetry[14,19]. This is verified by the transport measurements below.

## AHE in the Ru$_{0.8}$Cr$_{0.2}$O$_2$ (110) film

The longitudinal resistivity and Hall conductivity in Ru$_{0.8}$Cr$_{0.2}$O$_2$ (110) film were measured with currents along two in-plane directions, [001] and [1$\bar{1}$0], as shown in Supplementary Fig. 5 (see "Methods"). Both directions show a metallic state, and the anomalous Hall conductivity (AHC) measured with the current along [1$\bar{1}$0] exhibits a larger signal. Therefore, we below present the results of the AHE with the current along [1$\bar{1}$0].

Figure 4a shows the Hall conductivity (σ$_{xy}$) with a magnetic field sweeping at 3 K. Distinctly, a hysteretic feature is observed, in stark contrast to the absence of a hysteretic behavior in the M–H curve (Fig. 2c). This behavior demonstrates that the finite Hall vector is involved in the Ru$_{0.8}$Cr$_{0.2}$O$_2$ (110) film, even though the net magnetization is vanishingly small within the experimental accuracy. Thus, in the magnetic field range in which σ$_{xy}$ shows hysteretic behavior, one should take into account the coexistence of the two magnetic domains with opposite Hall vectors (i.e., the AHCs with opposite signs).

In general, the origin of σ$_{xy}$ consists of the external magnetic field (or ordinary Hall conductivity, σ$_{xy}^{OHE}$, proportional to H with a coefficient $k_o$) and the magnetism (or anomalous Hall conductivity, σ$_{xy}^{AHE}$). The σ$_{xy}^{AHE}$ is often dictated by the contribution proportional to the net magnetization, but in the present system, the antiferromagnetic order coupled with the special lattice symmetry can also contribute[3,6,14]. Thus, the observed σ$_{xy}$ can be described as the sum of the three components:

$$\sigma_{xy}(H) = \sigma_{xy}^{OHE}(H) + \sigma_{xy}^{M}(H) + \sigma_{xy}^{AF}(H) = k_o \cdot H + k_m \cdot M(H) + \sigma_{xy}^{AF}(H),$$
(1)

where σ$_{xy}^{M}$ is the anomalous Hall conductivity proportional to the field-induced net magnetic moment M with a coefficient $k_m$, and σ$_{xy}^{AF}$ is the anomalous Hall conductivity arising from the antiferromagnetic ordering[6]. Note that in the present field range, the magnetic field-dependent σ$_{xy}^{AF}$(H) is caused by the change in the relative volume of the two types of antiferromagnetic domains with opposite signs of AHC.

At sufficiently high magnetic fields, the hysteretic behavior disappears, and therefore, a single antiferromagnetic domain is expected. Thus, σ$_{xy}^{AF}$ is considered to be a constant, σ$_{xy}^{AF,0}$, at a sufficiently high magnetic field[14]. Utilizing the data of Hall conductivity and magnetization at 4–7 T, where the hysteretic behavior is absent, we can thus obtain the coefficients, $k_o$ and $k_m$, and σ$_{xy}^{AF,0}$. For clarity, by subtracting σ$_{xy}^{OHE}$ = $k_o \cdot H$, we display the experimental σ$_{xy}^{AHE}$ together with the fitting curve $k_m \cdot M$ + σ$_{xy}^{AF,0}$ as a function of the net magnetization in Fig. 4b. The value of σ$_{xy}^{AF,0}$ is ≈3.2 S/cm, which is indicated by the intercept of the fitting curve at M = 0. In the low-field region, the experimental σ$_{xy}^{AHE}$(H) deviates from the linear fitting. In the present framework, this deviation is attributable to the coexistence of two antiferromagnetic domains with opposite signs of AHC.

The evolutions of σ$_{xy}^{AF}$ and σ$_{xy}^{M}$ with magnetic field sweeping at 3 K are shown in Fig. 4c, where σ$_{xy}^{M}$ is set to $k_m \cdot M$, and σ$_{xy}^{AF}$ is obtained by subtracting σ$_{xy}^{M}$ from σ$_{xy}^{AHE}$. Interestingly, the σ$_{xy}^{AF}$ shows a hysteretic profile and a clear remnant value even at the vanishing net moment (Fig. 2c). Such features indicate an AHC contributed by the antiferromagnetic ordering, not due to the canting moment. The emergent σ$_{xy}^{AF}$ decreases as the temperature increases and disappears at 40–50 K (Fig. 4d and Supplementary Fig. 7), indicating the antiferromagnetic order transition point (T$_N$).

To gain further insight into the microscopic mechanisms of the σ$_{xy}^{AF}$ and σ$_{xy}^{M}$, we compared the AHC–σ$_{xx}$ scaling curves[2,36–39] among Ru$_{0.8}$Cr$_{0.2}$O$_2$ (110) films with different σ$_{xx}$, which was tuned by tailoring the thickness. As shown in Supplementary Fig. 8, all films are located at the crossover from dirty to intermediate regimes with 10$^3$ <σ$_{xx}$< 10$^4$ S/cm, thereby ruling out the skew scattering contribution, which is generally considered in high conductive metals (σ$_{xx}$ > 10$^6$ S/cm). Besides, a further analysis based on the σ$_{xy}^{M}$(T)–σ$_{xx}$(T)$^2$ profile gives an intrinsic Berry curvature term of 14 S/cm (Supplementary Note 3) and the extrinsic side-jump contribution of ~10 S/cm. These results indicate that the Berry curvature and extrinsic scattering microscopic mechanisms both contributes to σ$_{xy}^{M}$(T) in our films. We note that the σ$_{xy}^{M}$ value is similar to the AHC in ferromagnetic SrRuO$_3$ films grown by PLD, although the canting moment (0.04 μ$_B$/f.u.) of our Ru$_{0.8}$Cr$_{0.2}$O$_2$ (110) film is ~40 times smaller than the ferromagnetic moment in SrRuO$_3$ films[40,41]. We also note that the value of σ$_{xy}^{AF}$ in Ru$_{0.8}$Cr$_{0.2}$O$_2$ is one order of magnitude larger than the recently reported collinear antiferromagnetic semiconductor MnTe[16].

## Orientation-anisotropic anomalous Hall response

Finally, we show that the transport properties in our Ru$_{0.8}$Cr$_{0.2}$O$_2$ film also indicate the Hall vector along [110]. To address this issue

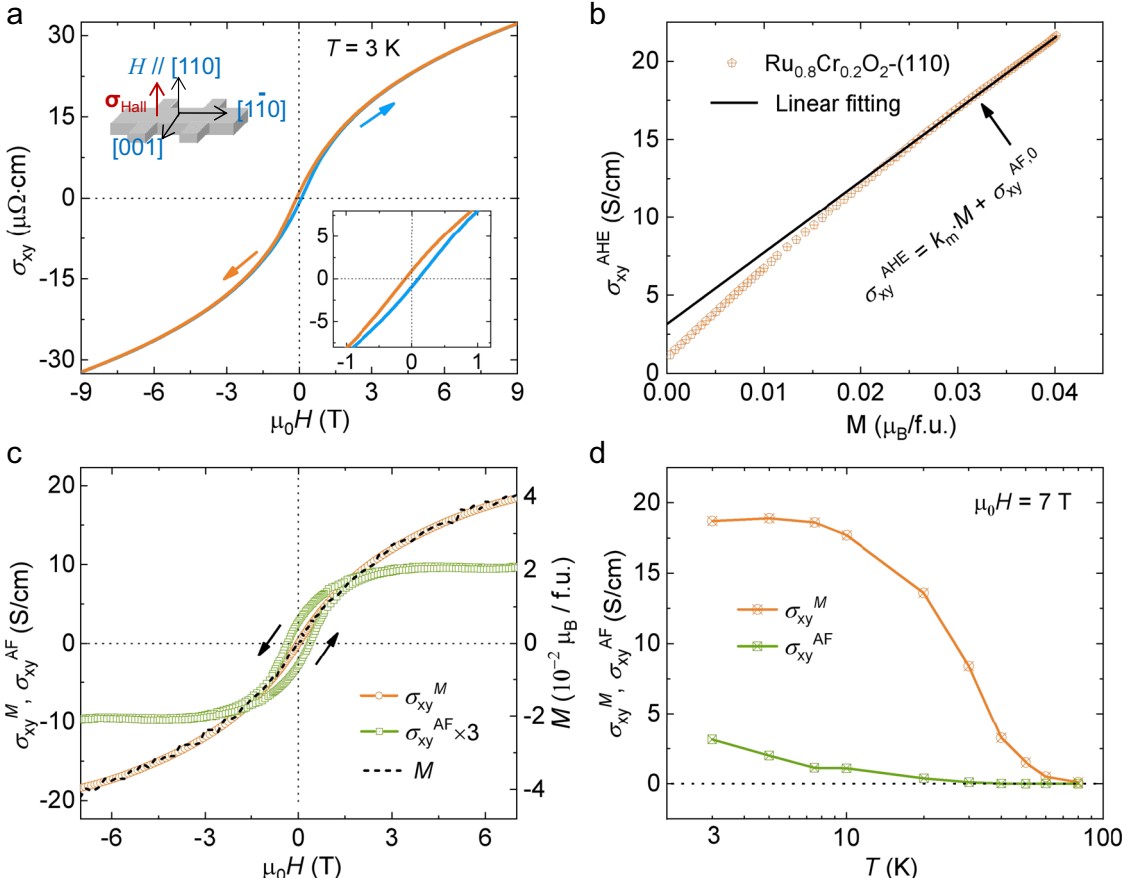

**Fig. 4 | Transport properties of the Ru₀.₈Cr₀.₂O₂ film grown on TiO₂ (110). a** Hall conductivity with magnetic field dependence at 3 K. Insets show the Hall configuration (left) and an expanded view of the low-field region (right). $\sigma_{Hall}$, Hall vector. **b** Anomalous Hall conductivity at 3 K with a dependence on the magnetic moment ($M$). The $\sigma_{xy}^{AHE}$ was obtained by subtracting a field-linear-dependent ordinary Hall contribution from $\sigma_{xy}$. $M$ was measured by an MPMS at 3 K. **c** Anomalous Hall conductivity derived from the canting moment (i.e., $\sigma_{xy}^{M}$) and the anti-ferromagnetic domain (i.e., $\sigma_{xy}^{AF}$) in Ru₀.₈Cr₀.₂O₂ (110) with a dependence on magnetic field sweeping at 3 K. The magnetic moment is shown by a dashed line. **d** Temperature-dependent $\sigma_{xy}^{M}$ and $\sigma_{xy}^{AF}$. The data at 7 T are used.

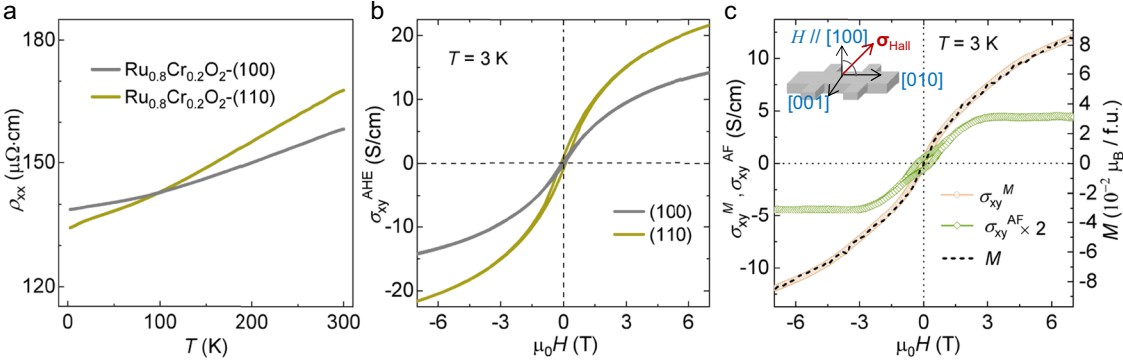

**Fig. 5 | Comparison of the transport behaviors for films grown along (100) and (110). a** Temperature-dependent longitudinal resistivities of the two films. **b** Magnetic field-dependent $\sigma_{xy}^{AHE}$ at 3 K for the two films. **c** $\sigma_{xy}^{M}$ and $\sigma_{xy}^{AF}$ with magnetic-field dependence at 3 K for the film grown along the (100) orientation. The magnetic moment is shown by a dashed line. During the transport measurement on the Ru₀.₈Cr₀.₂O₂-(100) film, the current was applied along the [010] direction with a Hall voltage along the [001] direction for comparison. Inset, the illustration of the Hall bar and the Hall vector ($\sigma_{Hall}$).

experimentally, we referred to the fact that the transverse anomalous Hall current ($J_H$) is given by $J_H = E \times \sigma_{Hall}$[14,18], where $E$ represents the applied external electric field, and carried out transport measurements on another film grown on TiO₂ (100). Herein, the current was applied along the [010] direction to keep the Hall voltage also along the [001] direction for comparison. The temperature dependence of AHE is similar to that observed for the [110]-oriented films (Supplementary Fig. 7), indicating that the transition temperature is not affected by the

orientation of the substrate. As shown in Fig. 5a, b, the longitudinal conductivities at low temperatures of the two films are very close to each other, while the $\sigma_{xy}^{AHE}$ that emerges from the (100) film is distinctly smaller than the value for the film grown on TiO₂ (110). Upon further analyzing the magnetization and the anomalous Hall contributions of $\sigma_{xy}^{M}$ and $\sigma_{xy}^{AF}$, as shown in Fig. 5c, we find that both of the anomalous Hall components are suppressed compared with those in Fig. 4c. Furthermore, we find that the saturated $\sigma_{xy}^{AF}$ in the

[100]-oriented film is approximately ~2.2 S/cm, which is 0.7 ($\simeq \sin 45°$) times that in the [110]-oriented sample, ~3.2 S/cm. These transport results further support that the Hall vector is directed along the [110] direction in this compound, as illustrated in Fig. 5c, inset.

## Discussion

To support the assumed collinearity of the antiferromagnetism, it may be useful to refer to the fact that the noncollinear antiferromagnetic materials with frustrated spin interactions are prone to show a large value of $|\theta_W/T_N|$ (>10)[42]. In our $Ru_{0.8}Cr_{0.2}O_2$ film, the value of $|\theta_W/T_N|$ is found to be small, 1.5–1.8, which at least does not contradict the assumed collinearity. While our experimental observations in the $Ru_{0.8}Cr_{0.2}O_2$ films can be consistently understood by assuming that the collinear antiferromagnetism with the Néel vector along [110] is realized, more rigorous evidence of the magnetic structure, for instance using neutron diffraction, remains a challenge in the future studies.

In summary, by inducing Fermi-level shift, we have succeeded in changing the easy axis of the Néel vector from that of $RuO_2$ and thus observing the zero-field AHE in the reconstructed collinear antiferromagnetic rutile metal, $Ru_{0.8}Cr_{0.2}O_2$. While the antiferromagnetic metallic phase is rare in correlated oxides[31,32,43], our study indicates a possibility to broaden the candidate materials and produce variations of the antiferromagnetic Néel vector by doping. We envision that this material-design strategy may also work and be helpful to explore the AHE in other rutile oxides, such as $ReO_2$[44].

## Methods

### DFT calculations and Wannierization
We computed the Bloch wavefunctions for $RuO_2$ on the basis of density functional theory (DFT) using the Quantum ESPRESSO package[45,46]. We first assumed a nonmagnetic structure without spin-orbit coupling and used the projector augmented wave pseudopotential[47] and the generalized gradient approximation of the Perdew–Burke–Ernzerhof exchange correlation functional[48]. We used lattice constants of a = 4.492 Å and c = 3.107 Å. The energy cutoff for the wave function and the charge density, $e_{wfc}$ and $e_{rho}$, respectively, were set to $e_{wfc}$ = 60 Ry and $e_{rho}$ = 400 Ry. We used $\boldsymbol{k}$-point meshes of $12 \times 12 \times 16$ and $16 \times 16 \times 16$ in the self-consistent field (scf) and non-scf calculations, respectively. After the DFT calculations, Wannierization was performed by using the wannier90 package[49,50], in which the Bloch orbitals were projected onto the $t_{2g}$ orbitals of Ru ions with $16 \times 16 \times 16$ $\boldsymbol{k}$-point grids.

To calculate the electronic states of $Ru_{1-x}Cr_xO_2$, with x = 0, 0.25, and 0.5, we replaced the Ru-sites denoted as Ru-1 or Ru-2 in Supplementary Fig. 3a with Cr. In this calculation, we set $e_{rho}$ = 500 Ry, and the spin-orbit coupling was not included. For the x = 0 and 0.5 systems, we took $24 \times 24 \times 32$ $\boldsymbol{k}$-mesh for the scf calculation. When we calculated the ground states of $Ru_{0.75}Cr_{0.25}O_2$, we used the supercell with the b- or c-axis doubled. We took the $\boldsymbol{k}$-mesh of $24 \times 12 \times 32$ ($24 \times 24 \times 16$) when the b- (c-)axis was doubled for the scf calculation. We found that the supercell with the b-axis doubled was more energetically stable, which we have used for discussion. To obtain the projected density of states (PDOS) of the x = 0 and 0.5 systems, we performed the non-scf calculations with $24 \times 24 \times 32$ $\boldsymbol{k}$-mesh after the scf calculation and then calculated the PDOS. We also calculated the PDOS of $RuO_2$ with the DFT + $U$ method with $U$ = 3 eV and nonmagnetic $Ru_{1-x}Cr_xO_2$ with x = 0 and 0.5, where we set $e_{rho}$ = 500 Ry and took $24 \times 24 \times 32$ $\boldsymbol{k}$-points for the scf and non-scf calculations.

For examining the orientation of the Néel vector, we performed the DFT + $U$ calculation for $RuO_2$ with the spin-orbit coupling for the three cases where the Néel vector was initially along [001], [100], and [110]. We took $U$ = 3 eV. We used $24 \times 24 \times 32$ $\boldsymbol{k}$-points and set $e_{rho}$ = 500 Ry. The convergence threshold for the calculation of the Néel vector orientation was set as $10^{-6}$ Ry.

### DMFT calculations
The Wannier functions obtained above define a tight-binding model for the three Ru $t_{2g}$ orbitals of $RuO_2$. Using this as the one-body part of the Hamiltonian, we constructed a multiorbital Hubbard model with intra(inter)orbital Coulomb interaction $U(U')$ and Hund's coupling and pair hopping $J$. We solved the model within the dynamical mean field theory (DMFT)[51] at zero temperature. As a solver for the DMFT impurity problem, we used the exact diagonalization method[52], where the dynamical mean field was represented by nine bath sites. To obtain the antiferromagnetic solution, we assumed opposite spin polarizations at neighboring Ru sites in the unit cell. For the interaction parameters, we assumed $U = U' + 2J$ and $J = U/5$ for the sake of simplicity.

### Thin-film growth, X-ray diffraction, and XAS
The $Ru_{1-x}Cr_xO_2$ films were grown on the rutile $TiO_2$ substrate by the PLD method with stoichiometric targets. During sample growth, the substrate temperature was kept at 290 °C to suppress interfacial diffusion, and the oxygen partial pressure was kept at 20 mTorr. The laser fluence was 1.2 J/$cm^2$ (KrF, $\lambda$ = 248 nm), and the deposition frequency was 3 Hz. After deposition, the samples were cooled to room temperature at a rate of 10 °C/min under an oxygen pressure of 10 Torr. The film thickness was determined directly with an X-ray reflectivity measurement. X-ray diffraction measurements were performed using a high-resolution diffractometer (Rigaku) with monochromatic Cu $K_{\alpha 1}$ ($\lambda$ = 1.5406 Å) X-rays. The stoichiometry in the thin film was checked by energy dispersive X-ray (EDX), and the ratio of Ru/Cr was confirmed to be very close to the target. The XAS curves of Cr L-edge were measured with a total electron mode, at 20 K, in beamline BL07U of Shanghai Synchrotron Radiation Facility.

### Transport and magnetization measurements
All of the electrical transport was carried out on Hall bar devices with a size of 300 μm × 60 μm, which were fabricated by photolithography. The milling process was carried out with $Ar/O_2$ (10:1) mixed ions and at a low speed to avoid oxygen vacancy formation on the $TiO_2$ surface. The transport measurements were carried out with a PPMS system (Quantum Design) with an in-plane DC current. The magnetoresistivity (MR) and its anisotropy were very small, as shown in Supplementary Fig. 9. The Hall conductivity $\sigma_{xy}$ was calculated as $\sigma_{xy} = -\rho_{yx}/(\rho_{xx}^2)$. The magnetization was measured using an MPMS system (Quantum Design) and obtained by subtracting the contribution from the $TiO_2$ substrate. The Hall vector ($\boldsymbol{\sigma}_{Hall}$) is defined as that in ref. 14.

## Data availability
All data used to generate the figures in the manuscript and supplementary information is available on Zenodo at: https://zenodo.org/record/8412950.

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

## Acknowledgements

This research was supported by JSPS KAKENHI (Grant Nos. 21H04437, 19H05825, 19H02594, 21H04442, 21K14398) and JST CREST (Grant No. JPMJCR1874). P.Y. was financially supported by the National Key R&D Program of China (grant No. 2021YFE0107900) and the National Natural Science Foundation of China (grant No. 52025024). K.D. was supported by the National Natural Science Foundation of China (Grant No. 12004159), Guangdong Basic and Applied Basic Research Foundation (Grant Nos. 2022A1515011915, 2019A1515110712) and the Shenzhen Science and Technology Program (Grant No. RCBS20210706092218039).

## Author contributions

M.W. and F.K. conceived the project. M.W. grew the thin films and performed the transport measurements with help from S. Shen. K.T. and S. Sakai performed the calculations with the supervision of R.A. M.W., Y.L., and D.T. conducted the XRD and magnetization measurements with support from P.Y. K.D. and C.L. conducted the XAS measurements. Z.W. and N.O. provided insights about the antiferromagnetic symmetry. All of the authors discussed the results and provided feedback.

## Competing interests

The authors declare no competing interests.
