## [Peer Review File · Nature Communications]

Reviewers' Comments:

Reviewer #1:

Remarks to the Author:

The paper by Meng Wang et al. presented an important advancement in enhancing the anomalous Hall effect (AHE) of rutile antiferromagnetic metal RuO₂. By simply inducing the Cr doping to the RuO₂/TiO₂(110) film, a large spontaneous anomalous Hall conductivity, comparable with the ferromagnetic SrRuO₃, was achieved in the antiferromagnetic Ru_{0.8}Cr_{0.2}O₂. Using various experimental and theoretical tools, the author attributes this large AHE to the Cr doping-induced enhancement of electron correlation and on-site moment. The overall quality and novelty of this manuscript is high. I recommend this work be published in Nature Communications after addressing the following points:

1. My first concern is the model for DFT calculation in this manuscript could be oversimplified. The DFT calculation is based on pure RuO₂, and the Cr doping effect is simply included by enlarging the Coulomb repulsion U . In this model, the Cr doping-induced lattice distortion, exchange interactions, and charge transfer between Cr and Ru sites, are neglected. I recommend enlarging the supercell and including the actual Cr dopants in the DFT calculation.
2. The current XAS results are inadequate to fully support the charge transfer picture. Additional XAS characterizations on the Ru_{1-x}Cr_xO₂ films with $x = 0.1$ and 0.3 are required.
3. The authors employed an ingenious method to separate the σ_{xy}^{AF} and σ_{xy}^M . I am curious whether both terms are intrinsic (originate from the non-vanishing Berry curvature). Considering that the magnetic dopants may also enhance the scattering-related extrinsic AHE, I wonder whether the AHE in Ru_{0.8}Cr_{0.2}O₂ also has a mixed origin. I recommend the authors further analyze the scaling between σ_{xy} and σ_{xx} , which can help to identify the detailed origin of these two AHE terms.
4. In Pages 8~9, the author attributed the different σ_{xy} measured along the (110) and (001)-oriented films to the differences in Berry curvatures. As mentioned in comment 2, the AHE could have a mixed origin, thus this point should be reconsidered. I recommend verifying this point by DFT calculations on the Berry curvatures.
5. In Page 9, line 239, the author mentioned: "...indicating that the effective magnetic field due to the antiferromagnetic ordering is likely aligned along the [110] direction in this compound." This statement is unclear to me. Does the author mean the Néel vector or the canted moments are preferentially aligned along [110]? I recommend the author clarify this point further.

Reviewer #2:

Remarks to the Author:

In this work the authors report experimental observation of the anomalous Hall effect in antiferromagnetic RuO₂ doped with Ir. Unlike in the previous experiments on pure RuO₂ they observe the AHE even at zero fields, which is referred here as the "spontaneous anomalous Hall effect". The AHE in antiferromagnetic systems is an intensively researched topic and this manuscript is an interesting addition to this field. However, I'm not convinced the conclusions of the manuscript are as substantiated as the authors claim and there are a number of issues with the manuscript. I cannot recommend the manuscript to be published in its present form.

1. A crucial aspect of the AHE in RuO₂ that is not discussed by the authors at all is that the AHE in RuO₂ in fact vanishes when the magnetic moments are along the c-axis, which is the orientation of magnetic moments found by the neutron diffraction experiments [13]. That is, when the magnetic moments are oriented along the easy axis, there is no AHE allowed by symmetry. When they are tilted from this direction the AHE becomes allowed by symmetry. Therefore, if this is indeed the correct magnetic order then the lack of hysteresis in the observed Hall signal in a previous experiment (Ref. 16) is the expected result and this is the explanation authors of Ref. 16 give for their observation.

This also means that if the effect of the Ir doping was, as the authors claim, only the increase in the local moment, then hysteresis would still not be expected. The fact that it is observed thus necessarily means that something else is going on. This needs to be discussed by the authors.

2. Even ignoring the previous point, I don't find the evidence the authors give for their interpretation of the experiments to be very strong. It seems reasonable that the doping by Ir could increase the Ru magnetic moment, but I'm not sure if the experimental results clearly show that this is indeed the case and I'm also not convinced that this is necessarily the explanation of the observed experimental difference to the pure RuO₂ results.

The experimental determination of the Ru magnetic moment comes from fitting to the Curie-Weiss law. It is not clear to me how this fitting is done and how the sublattice moment is obtained from the fit. Perhaps, this is well known, but the reference the authors give does not explain the procedure either and I'm sure other readers will also be confused by this so it would be useful if the authors could explain this more. Furthermore, I'm not convinced that this is an accurate measure of the sublattice moment. The Curie-Weiss law is empirical and at the very least I would assume that such determination of magnetic moment will carry significant uncertainty. Since this is the only experimental evidence of the magnetic moments change with the doping, which is the central point of the paper, this deserves more attention.

I'm also not so convinced by the previous experiment showing only a very tiny moment on the Ru atom. Although I have no specific reasons to doubt the accuracy of that experiment, this result is unexpected and very strange considering the large Néel temperature. I also note that even in the manuscript that reports this tiny moment, the DFT calculations have used much larger moments. Previous detailed theoretical study of the magnetic order in RuO₂ have also used much larger magnetic moments [PRB 99, 184432 (2019)].

Thus overall, I find the claim that the effect of Ir doping on the AHE is due to the enhanced moments to be quite weak. There is no strong evidence for this and it is not compatible with the symmetry of RuO₂.

3. I also don't think it's clear that the magnetic order will necessarily be unchanged from the pure RuO₂ (and it's questionable how well is the magnetic order in pure RuO₂ understood). It is reasonable to assume that these systems are indeed antiferromagnetic (except maybe the $x=0.3$ case), but the quite large doping by Cr could easily change the magnetic structure. This is especially important considering the symmetry discrepancy discussed above. In fact, I'm not sure if the authors can even make the claim that their samples are collinear antiferromagnets.

4. The authors claim that the $x=0.3$ case is likely ferrimagnetic. I don't understand why that should necessarily be the case. In systems where the AHE is allowed by symmetry, the symmetry also allows existence of a net magnetic moment, since the symmetry requirements for AHE and a net moment are the same. Such a moment may be small or even zero and is not necessarily related to the AHE but is always allowed by symmetry. Thus, if the zero field AHE is observed it is not surprising that a net moment is also observed.

5. I'm not sure if the term "spontaneous AHE" will be clear to readers. To me it seems strange and misleading. I understand what the authors mean by it: the fact that the AHE shows hysteresis and is thus observed even at zero field. Yet, as far as I can see such term has not been commonly used in the previous literature. Rather the term "spontaneous Hall effect" has been historically used as another name for the "anomalous Hall effect". This is not used much anymore, though occasionally it is still used. Thus, I think this may confuse readers since "spontaneous Hall effect" and "anomalous Hall effect" have historically meant the same thing. Furthermore, the term "spontaneous AHE" might imply that the regular AHE is not spontaneous, which is certainly not the case in general. To me it's also a strange expression because I would argue that even in previous experiments on RuO₂ the observed AHE is spontaneous. Although magnetic field is needed to observe it, this is only because the magnetic moments need to be tilted out of the c-axis. The origin of the effect still is a spontaneous symmetry breaking due to the magnetic order and the tilting could in principle be achieved by other means.

6. There are quite significant errors throughout the manuscript, mainly in the introduction, regarding the symmetry. The authors claim that the existence of AHE is determined by whether the material has a time-reversal symmetry. That is not correct. It is true that the AHE cannot exist

in materials with time-reversal symmetry, however, time-reversal symmetry is broken by any magnetic order including antiferromagnetic orders even those that do not allow the AHE. It is not true that antiferromagnets invariant under time-reversal followed by lattice translation have time-reversal symmetry. Rather such systems have a symmetry operation that is composed of time-reversal plus translation. Such symmetry operation is distinct from the time-reversal, but like the time-reversal it prohibits the existence of the AHE. There can be other symmetry operations that prohibit the AHE, in particular the time-reversal plus spatial inversion. Even symmetry operations that have nothing to do with time-reversal can prohibit AHE, for example, no AHE can exist in the presence of two rotation axes.

In RuO₂ the time-reversal plus translation and the time-reversal plus inversion symmetries are broken, but when the magnetic moments are along c axis, the AHE is still prohibited by symmetry, due to existence of other symmetry operations. When the magnetic moments are moved away from the c-axis then the AHE is allowed by symmetry.

7. The authors explain the existence of AHE and its symmetry properties using the Berry curvature. This is commonly done, but this should only be seen as illustration of how AHE can originate. There are other mechanisms beyond the Berry curvature and the existence of AHE is determined by symmetry arguments that are independent of the Berry curvature.

Furthermore, the authors claim on lines 206 and 227, that the origin of the observed AHE is the Berry curvature. As far as I can see, there is no basis for such a claim, and it is entirely possible that the observed AHE has an extrinsic origin in their case.

8. The authors claim that the distinguishing feature of their experiments compared to the previous experiments where hysteretic AHE in antiferromagnets was observed, is that the previous experiments used antiferromagnets with geometrical frustration.

My understanding is that at least in the case of Mn₃Sn or similar antiferromagnets there is in fact no geometrical frustration. Frustration would exist in these systems if the magnetic moments were collinear, but the non-collinear order that is observed is not frustrated.

Perhaps more importantly, even if there was a geometrical frustration, the manuscript suggests that this frustration is somehow crucial for the existence of the AHE, which is not the case. I also find the claim that the RuO₂ is a "much simpler" system than the previously studied antiferromagnets dubious. Simplicity can of course mean different things, but the Mn₃Sn antiferromagnet, for example, also has quite a simple magnetic and crystal structure.

I also point out that recently there has been observation of AHE in MnTe [PRL 130, 036702 (2023)], which is a collinear antiferromagnet similar to RuO₂ that certainly has no frustration. This work needs to be cited.

9. Is there a reason why the experiments did not include the case of pure RuO₂? It would be interesting to have this for comparison.

We appreciate the constructive comments and suggestions from both reviewers. Accordingly, we have carried out more experiments, calculations, and analyses as shown in the point-to-point response letter and revised manuscript (highlighted in purple color), which have greatly improved our manuscript.

Reviewer #1 :

The paper by Meng Wang et al. presented an important advancement in enhancing the anomalous Hall effect (AHE) of rutile antiferromagnetic metal RuO₂. By simply inducing the Cr doping to the RuO₂/TiO₂ (110) film, a large spontaneous anomalous Hall conductivity, comparable with the ferromagnetic SrRuO₃, was achieved in the antiferromagnetic Ru_{0.8}Cr_{0.2}O₂. Using various experimental and theoretical tools, the author attributes this large AHE to the Cr doping-induced enhancement of electron correlation and on-site moment. **The overall quality and novelty of this manuscript is high. I recommend this work be published in Nature Communications after addressing the following points:**

Response:

We thank the reviewer for the positive feedback as well as constructive suggestions. Below are our point-to-point responses to the reviewer's comments.

Comment 1: My first concern is the model for DFT calculation in this manuscript could be oversimplified. The DFT calculation is based on pure RuO₂, and the Cr doping effect is simply included by enlarging the Coulomb repulsion U . In this model, the Cr doping-induced lattice distortion, exchange interactions, and charge transfer between Cr and Ru sites, are neglected. I recommend enlarging the supercell and including the actual Cr dopants in the DFT calculation.

Response:

According to the reviewer's suggestions, we have further calculated the magnetic ground state of Ru_{1-x}Cr_xO₂ for $x = 0.25$ and 0.5 with $U = 0$. There are two crystallographically inequivalent sites in the unit cell of RuO₂, labeled as Ru-1 and Ru-2 (Fig. R1a), some of which we replaced with Cr in the calculations. For the calculation of Ru_{0.75}Cr_{0.25}O₂, we used the b -axis- or c -axis-doubled cell. We found that the b -axis-doubled cell has a smaller energy and thus used this for the calculation presented below.

As shown in Table R1, the magnetic ground state shows a ferrimagnetic order for both $x = 0.25$ (a) and 0.5 (b), indicating that the coupling between the nearest neighboring Cr and Ru ions is antiferromagnetic while the magnitudes of the two local moments are unequal. Moreover, we find no clear magnetic ordering for $x = 0$ due to weak electron correlation. In Fig. R1b, we compare the x -dependence of the local on-site moment of Ru ions, demonstrating that the pure RuO_2 shows a vanishingly small moment, whereas the moment of Ru emerges and increases gradually as the Cr doping level increases. These results clearly indicate that an antiparallel spin coupling evolves as a result of the charge transfer in the Cr-doped RuO_2 . Although the DFT calculation tells that the ferrimagnetic ground state would be observed if the Cr dopants were long-range ordered, the Cr ions in the actual experiments should be randomly distributed into the two sublattices. Thus, the DFT calculation indicates that the antiferromagnetic state with vanishingly small uniform moments is the most likely ground state for the randomly Cr-doped RuO_2 . This conclusion is consistent with the experimental observations for the lightly doped region ($x \leq 0.2$), strongly supporting our working hypothesis for the material design. These results have been added to Supplementary Information in the revised manuscript.

Fig. R1. DFT Calculations of $\text{Ru}_{1-x}\text{Cr}_x\text{O}_2$ lattice. a, Crystal structure of the unit cell of RuO_2 . The two crystallographically inequivalent sites of Ru are labeled. b, Calculated average Ru moment with a dependence on the Cr doping levels. The U is set as zero for all compositions.

Table R1. Calculated spin order and moments in $\text{Ru}_{0.75}\text{Cr}_{0.25}\text{O}_2$ with the b -axis doubled cell **(a)** and $\text{Ru}_{0.5}\text{Cr}_{0.5}\text{O}_2$ **(b)**.

a $\text{Ru}_{0.75}\text{Cr}_{0.25}\text{O}_2$				b $\text{Ru}_{0.5}\text{Cr}_{0.5}\text{O}_2$			
Cr at Ru-1 site		Cr at Ru-2 site		Cr at Ru-1 site		Cr at Ru-2 site	
Cr-1	1.8456 (μ_B)	Ru-1	0.1912 (μ_B)	Cr-1	1.8434 (μ_B)	Ru-1	0.4031 (μ_B)
Ru-2	-0.1911	Cr-2	-1.8456	Ru-2	-0.4023	Cr-2	-1.8436
Ru-1	0.0642	Ru-1	0.1912				
Ru-2	-0.1911	Ru-2	-0.0642				

Comment 2: The current XAS results are inadequate to fully support the charge transfer picture. Additional XAS characterizations on the $\text{Ru}_{1-x}\text{Cr}_x\text{O}_2$ films with $x = 0.1$ and 0.3 are required.

Response:

Following the reviewer's suggestion, we have further carried out the XAS measurements with a high-resolution synchrotron soft X-ray. As shown in Fig. R2, the Cr in all $\text{Ru}_{1-x}\text{Cr}_x\text{O}_2$ films exhibits an intermediate valence state between +3.25 and +3.5. The peaks of the materials show a gradual shift to lower energy as the Cr doping level increases from 0.1 to 0.3, indicating the gradual reduction of the valence. The observed tendency is consistent with the scenario of the charge transfer and resulting Fermi-level reconstruction in our experiments. We have updated the related figure in the revised manuscript.

Fig. R2. XAS of the $\text{Ru}_{1-x}\text{Cr}_x\text{O}_2$ films around the Cr L-edges. The spectra of $\text{Cr}^{+3.25}$ and $\text{Cr}^{+3.5}$ are obtained from the reference with a $\text{La}_{0.75}\text{Sr}_{0.25}\text{CrO}_3$ and $\text{La}_{0.5}\text{Sr}_{0.5}\text{CrO}_3$, respectively.^[1]

Comment 3: The authors employed an ingenious method to separate the σ_{xy}^{AF} and σ_{xy}^M . I am curious whether both terms are intrinsic (originate from the non-vanishing Berry curvature). Considering that the magnetic dopants may also enhance the scattering-related extrinsic AHE, I wonder whether the AHE in Ru_{0.8}Cr_{0.2}O₂ also has a mixed origin. I recommend the authors further analyze the scaling between σ_{xy} and σ_{xx} , which can help to identify the detailed origin of these two AHE terms.

Response:

Prior theoretical works have shown the anomalous Hall conductivity σ_{xy}^{AHE} depends on the longitudinal conductivity σ_{xx} in ferromagnetic materials,^{[2],[3]} which has identified the following three regimes: (i) the clean limit with σ_{xx0} (σ_{xx} at 0 K) $> 10^6$ S/cm, and $\sigma_{xy}^{AHE} \propto \sigma_{xx0}$ mainly due to the extrinsic skew scattering; (ii) the intermediate regime for σ_{xx} of $\sim 10^4$ – 10^5 S/cm, and the σ_{xy}^{AHE} is nearly constant mainly due to the intrinsic Berry curvature mechanism and the weak σ_{xx} dependence may appear due to the extrinsic side-jump mechanism; (iii) the dirty regime for $\sigma_{xx0} < 10^4$ S/cm, and $\sigma_{xy}^{AHE} \propto \sigma_{xx0}^{1.6}$ due to the suppressed intrinsic mechanism.

Fig. R3. Anomalous Hall conductivity as a function of the longitudinal conductivity in Ru_{0.8}Cr_{0.2}O₂. **a**, Thickness dependence of the R - T curves of the Ru_{0.8}Cr_{0.2}O₂ films. **b-e**, Fittings of the σ_{xy}^{AHE} - M curves at high-field regions to obtain the σ_{xy}^M and σ_{xy}^{AF} components for the films with a thickness of 40 nm (**b**), 20 nm (**c**), 10 nm (**d**), and 3 nm (**e**). **f**, Evolution of σ_{xy}^M and σ_{xy}^{AF} with the changing of σ_{xx0} . σ_{xx0} are set by the conductivity at 3 K. Dashed lines

separating the three regimes are taken from the reference paper.^[4] **g**, Linear fitting of the normalized $\sigma_{xy}^M - \sigma_{xx}(T)^2$ with the data of 3 - 20 K used for the 10 nm, 20 nm, and 40 nm films. The intercept denotes the intrinsic term of anomalous Hall conductivity (σ_{xy}^{int}). Inset, an enlarged view of the data.

Following the reviewer's suggestion, we have further studied the scaling of the σ_{xy}^{AHE} and σ_{xx} in our system. We tuned the longitudinal conductivity of $\text{Ru}_{0.8}\text{Cr}_{0.2}\text{O}_2$ films by tailoring the thickness. Figure R3a shows that as the film thickness changes from 40 nm to 3 nm, the itinerant metallic phase gradually changes into an electron-hopping state with an increased resistivity. We further analyzed the field-induced uniform moment and antiferromagnetic contributions (i.e., σ_{xy}^M and σ_{xy}^{AF} , respectively) from the $\sigma_{xy}^{\text{AHE}}-M$ curves, as shown in Fig. R3b-e, and the values of σ_{xy}^M and σ_{xy}^{AF} are plotted against σ_{xx0} (i.e. conductivity at 3 K) as shown in Fig. R3f. We note that all the films ($\sigma_{xx0} = 4-8 \times 10^3$ S/cm) are located at the crossover between the dirty regime and the intermediate regime, while a higher σ_{xx0} could not be achieved due to the film quality (defects density) limited by the PLD growth technique. The σ_{xy}^M and σ_{xy}^{AF} show a dramatic increase as the system moves away from the dirty limit, while they exhibit a gradually saturating tendency as the σ_{xx0} approaches 10^4 S/cm. Based on this behavior, we can first rule out the skew scattering contribution to σ_{xy}^{AHE} , which is usually considered only for the case of very clean materials with $\sigma_{xx0} > 10^6$ S/cm.

We then tried to distinguish the side-jump and intrinsic contributions of the magnetization induced σ_{xy}^M in the high conductivity region (i.e. 10 - 40 nm films), which is generally considered through the following equation:^{[5],[6]}

$$\sigma_{xy}^{\text{AHE}} = (\alpha \sigma_{xx0}^{-1} + \beta \sigma_{xx0}^{-2}) M \sigma_{xx}(T)^2 + b M, \quad (1)$$

where M is the magnetization, $[\alpha \sigma_{xx0}^{-1} M \sigma_{xx}(T)^2]$, $[\beta \sigma_{xx0}^{-2} M \sigma_{xx}(T)^2]$, and $(b M)$ correspond to the anomalous Hall conductivity due to skew scattering, side-jump, and intrinsic mechanisms, respectively. The skew scattering mechanism is negligible in our thin films; i.e., $\alpha \approx 0$. We then carried out the fitting of $\sigma_{xy}^M \sim \sigma_{xx}(T)^2$ for the 10, 20, and 40 nm thickness films, as shown in the Fig. R3g. Considering a condition that M does not decrease too much with increasing temperature, the data of 3 to 20 K were used. Note that the conductivity does not change significantly in this temperature range, too. From these analyses, we obtained the intrinsic and side-jump contributions to σ_{xy}^M , which originate from the uniform moment, are ~ 14 S/cm and ~ 10 S/cm,

respectively. On the other hand, we didn't find a model that can quantitatively distinguish the side-jump and intrinsic Berry curvature contributions to the collinear-antiferromagnetic-order-induced σ_{xy}^{AF} , the magnitude of which is ~ 3.2 S/cm. However, we think it is reasonable to expect that the relative magnitudes of the intrinsic Berry curvature and extrinsic side-jump contributions to σ_{xy}^{AF} are similar to the case of σ_{xy}^{M} .

We have added the related discussions to the main text and Supplementary Information.

Comment 4: In Pages 8~9, the author attributed the different σ_{xy} measured along the (110) and (001)-oriented films to the differences in Berry curvatures. As mentioned in comment 3, the AHE could have a mixed origin, thus this point should be reconsidered. I recommend verifying this point by DFT calculations on the Berry curvatures.

Response:

We thank the reviewer's comments suggesting DFT calculations to see the values of the Berry curvature, i.e., the antiferromagnetic anomalous Hall conductivity σ_{xy}^{AF} . However, we note that the σ_{xy}^{AF} of RuO₂ is very sensitive to the parameters for calculations, such as the Hubbard U and the Fermi energy, as shown in the Extended Data Fig. 1 in the reference paper [7]; namely, it is difficult to give a convincing estimation of σ_{xy}^{AF} by DFT.

Instead, by considering the Hall vector $\boldsymbol{\sigma}_{\text{Hall}} = (\sigma_{yz}, \sigma_{zx}, \sigma_{xy})$, below we show that the (110) and (001)-oriented films exhibit transport properties consistent with each other. The transverse anomalous Hall current (\mathbf{J}_{H}) induced by the collinear antiferromagnetic texture can be expressed as the cross product of the Hall vector and the applied electric field (\mathbf{E} , parallel to applied longitudinal current), i.e. $\mathbf{J}_{\text{H}} = \mathbf{E} \times \boldsymbol{\sigma}_{\text{Hall}}$, [8] where the direction of $\boldsymbol{\sigma}_{\text{Hall}}$ is determined by the antiferromagnetic Néel vector (\mathbf{L}) and the value is determined by the Berry curvature and extrinsic scattering. Here, we determined that the $\boldsymbol{\sigma}_{\text{Hall}}$ should be along [110] in our case as supported by the magnetic anisotropy measurements and calculations (for more details, please see our response to Reviewer #2's comment 1). Considering that the two films possess almost equal σ_{xx} and identical film quality, the $\boldsymbol{\sigma}_{\text{Hall}}$ should be identical. In fact, as detailed in our response to Comment 5, the suppressed anomalous Hall voltage in the

(110)-oriented film can be quantitatively explained in terms of the geometric view that E forms 90° or 45° with respect to [110] axis for the (110) and (100) films, respectively.

We have updated the related part in the revised manuscript.

Comment 5: In Page 9, line 239, the author mentioned: "...indicating that the effective magnetic field due to the antiferromagnetic ordering is likely aligned along the [110] direction in this compound." This statement is unclear to me. Does the author mean the Néel vector or the canted moments are preferentially aligned along [110]? I recommend the author clarify this point further.

Response:

Using the term "effective magnetic field" we were trying to describe a crystal plane perpendicular to it, where the Hall effect can be observed: in the previous study, instead of "effective magnetic field", the term "Hall vector" $\sigma_{\text{Hall}} = (\sigma_{yz}, \sigma_{zx}, \sigma_{xy})$ was used [8]. As you pointed out, this was a confusing expression and we corrected it.

Related to your comments, we confirmed that the Néel vector (L) in the $\text{Ru}_{0.8}\text{Cr}_{0.2}\text{O}_2$ film is along the [110] direction by the magnetic anisotropy measurements and first-principles calculations (for more details, please see our response to Reviewer #2's comment 1). The symmetry consideration concludes that when $L \parallel [110]$, the σ_{Hall} will also be along [110], and the anomalous Hall effect can be observed in the plane perpendicular to [110], consistent with our Hall measurements. Furthermore, when we measured the (100)-oriented sample, which forms 45° with respect to σ_{Hall} (Fig. R4, inset), we observed a suppressed anomalous Hall voltage, which is about $\sin 45^\circ$ times the Hall voltage observed in the (110)-oriented film. This observation is well explained by considering that the Hall vector is along the [110] direction in our thin films. We have further clarified this point in the revised manuscript.

Fig. R4. Magnetic field dependent σ_{xy}^{AF} and σ_{xy}^M measured in the $\text{Ru}_{0.8}\text{Cr}_{0.2}\text{O}_2$ [100]-oriented sample at 3 K. Inset, the illustration of the Hall bar and the Hall vector (σ_{Hall}).

Reviewer #2 :

In this work the authors report experimental observation of the anomalous Hall effect in antiferromagnetic RuO_2 doped with Cr. Unlike in the previous experiments on pure RuO_2 they observe the AHE even at zero fields, which is referred here as the “spontaneous anomalous Hall effect”. The AHE in antiferromagnetic systems is an intensively researched topic and this manuscript is an interesting addition to this field. However, I’m not convinced the conclusions of the manuscript are as substantiated as the authors claim and there are a number of issues with the manuscript. I cannot recommend the manuscript to be published in its present form.

Response:

We thank the reviewer for the critical comments on our work, which have helped us to further analyze the data, unveil the physics behind the observation, and clarify the related points. Below are our point-to-point responses to the reviewer’s comments.

Comment 1: A crucial aspect of the AHE in RuO_2 that is not discussed by the authors at all is that the AHE in RuO_2 in fact vanishes when the magnetic moments are along the c-axis, which is the orientation of magnetic moments found by the neutron diffraction experiments [13]. That is, when the magnetic moments are oriented along the easy axis, there is no AHE allowed by symmetry. When they are tilted from this direction the AHE becomes allowed by symmetry. Therefore, if this is indeed the correct magnetic order then the lack of hysteresis in the observed Hall signal in a previous experiment (Ref. 16) is the expected result and this is the explanation authors

of Ref. 16 give for their observation.

This also means that if the effect of the Cr doping was, as the authors claim, only the increase in the local moment, then hysteresis would still not be expected. The fact that it is observed thus necessarily means that something else is going on. This needs to be discussed by the authors.

Response:

We agree with the reviewer that the Hall vector $\boldsymbol{\sigma}_{\text{Hall}} = (\sigma_{yz}, \sigma_{zx}, \sigma_{xy})$, as defined in the reference paper [8], should vanish when the Néel vector is along the [001] direction. As detailed below, however, our magnetometry indicates that the Néel vector in our Cr-doped RuO₂ is along the [110] direction, different from the previous reports [7]. Thus, the magnetic space group (MSG) of our thin film is $Cmm'm'$, and this symmetry allows for a finite Hall vector along the [110] direction.

Before explaining our magnetometry results, let us review the symmetry argument for the case of RuO₂ with the Néel vector along the [001] direction (i.e., the magnetic space group MSG is $P4_2'mnm'$ as reported in the literature [8],[9]). The simplest argument would be to apply the symmetry operations of the corresponding magnetic point group (MPG) to the Hall vector $\boldsymbol{\sigma}_{\text{Hall}}$. The $P4_2'mnm'$ has C_{2x} and C_{2y} as the MPG symmetry operations: $C_{2x}(\sigma_{yz}, \sigma_{zx}, \sigma_{xy}) = (\sigma_{yz}, -\sigma_{zx}, -\sigma_{xy})$ and $C_{2y}(\sigma_{yz}, \sigma_{zx}, \sigma_{xy}) = (-\sigma_{yz}, \sigma_{zx}, -\sigma_{xy})$, demonstrating $\boldsymbol{\sigma}_{\text{Hall}} = \mathbf{0}$; i.e., σ_{yz} , σ_{zx} , and σ_{xy} are all prohibited. In contrast, when the Néel vector is along [100] and [110], the MSG is $Pnn'm'$ and $Cmm'm'$, respectively. The MPG symmetry operations in $Pnn'm'$ include $\mathcal{T}C_{2x}$, C_{2y} and $\mathcal{T}C_{2z}$, where \mathcal{T} represents the time-reversal operation; i.e., $\mathcal{T}C_{2x}(\sigma_{yz}, \sigma_{zx}, \sigma_{xy}) = (-\sigma_{yz}, \sigma_{zx}, \sigma_{xy})$ and $\mathcal{T}C_{2z}(\sigma_{yz}, \sigma_{zx}, \sigma_{xy}) = (\sigma_{yz}, \sigma_{zx}, -\sigma_{xy})$. Thus, only σ_{zx} can be allowed. Furthermore, given the fact that $Pnn'm'$ and $Cmm'm'$ consider different principal axes, one can conclude that the Hall vector is finite and directed along [010] (when the Néel vector \parallel [100]) and [110] (when the Néel vector \parallel [110]). Thus, AHE is allowed by symmetry when the applied magnetic field is not perpendicular to $\boldsymbol{\sigma}_{\text{Hall}}$.

Therefore, to observe the AHE in RuO₂, a situation in which the Néel vector is off the [001] direction is required. In the previous study [7], the authors concluded that the Néel vector is along the [001] direction and thus the external magnetic field was necessary to tilt the Néel vector from the [001] direction. This situation is clearly different from the case of our Cr-doped thin films.

In this revision, we discussed the direction of the Néel vector from both DFT

calculations and magnetometry. Figures R5a and R5b show the result of the DFT calculation on pure RuO₂ with $U = 3$ eV. We find that (i) the Néel vector along the [001] direction has the lowest energy, whereas the energy differences for the Néel vectors along [100] and [110] are very small (Fig. R5a), (ii) the Néel vector along the [100] direction is accompanied by a net magnetic moment of $0.08 \mu_B / \text{u.c.}$, which is inconsistent with our experiments and thus can be excluded from the candidate of the magnetic easy axis; and (iii) the magnetic anisotropy in RuO₂ is sensitive to the electron filling (i.e., the Fermi level) (Fig. R5b). Thus, the DFT calculation implies that the magnetic anisotropy of RuO₂ and Cr-doped RuO₂ is a subtle problem that should be resolved by experiment. Figure R5c shows the magnetization curves for $H \parallel [001]$ and $H \parallel [110]$. The field-induced magnetic moment along the [001] direction is obviously larger than that along the [110] direction. Given that in a collinear antiferromagnet, the smallest and largest field-induced magnetizations are generally observed when the Néel vector is parallel and perpendicular to the magnetic field, respectively ^[10]. Thus, the observations shown in Fig. R5c demonstrate that the Néel vector in Ru_{0.8}Cr_{0.2}O₂ is not along the [001] axis but most likely along the [110] axis. Thus, the MSG of our system should not be $P4_2'/mnm'$ but $Cmm'm'$, which allows for the observation of the zero-field AHE with hysteretic behavior. Furthermore, we also found that the magnitude of the AHE for the (100)-oriented sample is approximately $1/\sqrt{2}$ ($= \sin 45^\circ$) times that for the (110)-oriented sample (Fig. R4). This observation further supports our scenario that the Néel vector orientation in our Ru_{0.8}Cr_{0.2}O₂ thin film is along the [110] direction.

We appreciate the comments on the symmetry from the reviewer, and we have revised our manuscript based on the above discussions.

Fig. R5. Magnetic anisotropy and Néel vector orientation in $\text{Ru}_{0.8}\text{Cr}_{0.2}\text{O}_2$ film. **a**, Calculated energy (E) and net moment (M) in pure RuO_2 depending on the orientations of the Néel vector. The state with the Néel vector along [001] exhibits the lowest energy without spontaneous moment. The states with the Néel vector along [100] or [110] show about 5 meV higher energy, while the [100] case is accompanied by a spontaneous moment of 0.08 μ_B / u.c. **b**, Calculated energy difference of $E_{[110]} - E_{[001]}$ in pure RuO_2 depending on the Fermi level. The graph is reproduced from the reference paper [7]. The Cr-doping-induced charge transfer from Ru to Cr in our experiments corresponds to a shift of the Fermi level. **c**, Magnetization of $\text{Ru}_{0.8}\text{Cr}_{0.2}\text{O}_2$ measured under a magnetic field with a different orientation. The results of $H \parallel [110]$ and [001] were measured from the (110)-oriented sample with field along out-of-plane [110] and in-plane [001], respectively. Inset, illustration of the orientation of antiferromagnetic local moments at zero field.

Comment 2: Even ignoring the previous point, I don't find the evidence the authors give for their interpretation of the experiments to be very strong. It seems reasonable that the doping by Cr could increase the Ru magnetic moment, but I'm not sure if the experimental results clearly show that this is indeed the case and I'm also not convinced that this is necessarily the explanation of the observed experimental difference to the pure RuO_2 results.

The experimental determination of the Ru magnetic moment comes from fitting to the

Curie-Weiss law. It is not clear to me how this fitting is done and how the sublattice moment is obtained from the fit. Perhaps, this is well known, but the reference the authors give does not explain the procedure either and I'm sure other readers will also be confused by this so it would be useful if the authors could explain this more. Furthermore, I'm not convinced that this is an accurate measure of the sublattice moment. The Curie-Weiss law is empirical and at the very least I would assume that such determination of magnetic moment will carry significant uncertainty. Since this is the only experimental evidence of the magnetic moments change with the doping, which is the central point of the paper, this deserves more attention.

Response:

We thank the reviewer for these critical comments. We agree that the Curie-Weiss analysis is not smoking-gun evidence of the enhancement of the local magnetic moment. However, given that the detailed characterization of the magnetic properties of thin films is quite challenging, this is the best-effort experimental result that we can show at present. To compensate for this, in this revision, we performed further DFT calculations and found a tendency for the local magnetization to grow with Cr doping (for details, please see our response to Reviewer #1's comment 1 and Fig. R1).

Here, let us explain more details about the Curie-Weiss law, which is derived within the mean-field approximation and has been widely used for the characterization of the magnetic properties based on magnetic susceptibility^{[10],[11],[12]}. The Curie-Weiss law, which describes the relationship between the magnetic susceptibility (χ) and temperature (T) in the temperature regions above a magnetic ordering transition, has been widely used to see whether the dominant interaction is ferromagnetic or antiferromagnetic:

$$\chi = C / (T - \theta_w), \quad (2)$$

where C is the Curie constant and θ_w is often referred to as the Weiss temperature (or Weiss constant). The fitting of Equation (2) with a positive or negative θ_w indicates the ferromagnetic or antiferromagnetic interaction, respectively. Furthermore, the constant C is given by:

$$C = N_A \cdot \mu_0 \cdot \mu_{\text{eff}}^2 / (3k_B), \quad (3)$$

where $\mu_{\text{eff}}^2 = g_J(J+1)J \cdot \mu_B^2$, k_B is Boltzmann's constant, N_A is the Avogadro constant, g_J is the Landé g-factor, μ_B is the Bohr magneton, and J is the angular momentum quantum number. The Equation (3) calculated with the Gaussian units gives the

magnitude of the effective magnetic moment ^[11]:

$$\mu_{\text{eff}} = (8C)^{1/2} \mu_{\text{B}} \quad [\text{cgs}] \quad (4)$$

In our work, the $\chi^{-1}-T$ curves were fitted to obtain the constant C , and then the μ_{eff} was calculated from Equation (4) for three different doping levels. We agree that the μ_{eff} obtained from such a fitting may contain some deviation from the real value, however, our main purpose is to reveal an evolution tendency with the increase of the amount of the Cr doping, and the observed tendency is found to be well consistent with our DFT calculations shown in Fig. R1b.

We note the $\chi^{-1}-T$ fitting should be performed for the data above the magnetic transition temperature, while the RuO₂ is reported to have a T_{N} much higher than the room temperature,^[13] which makes the comparison with RuO₂ difficult in our measurements.

Comment 3: I'm also not so convinced by the previous experiment showing only a very tiny moment on the Ru atom. Although I have no specific reasons to doubt the accuracy of that experiment, this result is unexpected and very strange considering the large Néel temperature. I also note that even in the manuscript that reports this tiny moment, the DFT calculations have used much larger moments. Previous detailed theoretical study of the magnetic order in RuO₂ have also used much larger magnetic moments [PRB 99, 184432 (2019)].

Thus overall, I find the claim that the effect of Cr doping on the AHE is due to the enhanced moments to be quite weak. There is no strong evidence for this and it is not compatible with the symmetry of RuO₂.

Response:

We agree that the ultra-small local moment and the high transition temperature in RuO₂ are in stark contrast to the general tendency that a high transition temperature is accompanied by an appreciable local magnetic moment. We believe that the origin of such unconventional phenomena deserves further research. In this work, we mainly focus on the AHE originating from the collinear antiferromagnetic rutile even with vanishingly small net magnetization. We believe that our work is in stark contrast to the reported AHE in pure RuO₂ at high magnetic fields ^[7].

As discussed in our response to Comment-1 and 2, the Cr-doping-induced reconstruction of the Fermi level plays an important role in both increasing the local

moment and changing the direction of the Néel vector from [001] to [110]. We have revised the manuscript and the Supplementary Information with the above discussions.

Comment 4: I also don't think it's clear that the magnetic order will necessarily be unchanged from the pure RuO₂ (and it's questionable how well is the magnetic order in pure RuO₂ understood). It is reasonable to assume that these systems are indeed antiferromagnetic (except maybe the x=0.3 case), but the quite large doping by Cr could easily change the magnetic structure. This is especially important considering the symmetry discrepancy discussed above. In fact, I'm not sure if the authors can even make the claim that their samples are collinear antiferromagnets.

Response:

We thank the reviewer for this comment. We must admit that it is difficult to experimentally obtain smoking-gun evidence of the collinear antiferromagnetic order in our thin films. Nevertheless, we believe that the collinear antiferromagnetic state in Ru_{0.8}Cr_{0.2}O₂ is quite likely based on the following three points.

(1) We note that the Fermi energy and the lattice parameters of rutile CrO₂ and RuO₂ are very close to each other, while the t_{2g} orbital of the Cr 3d levels is a little lower than that of the Ru 4d levels. Thus, the Cr doping is inevitably accompanied by the charge transfer from Ru to Cr, facilitating an antiparallel coupling between the nearest neighboring ions, as illustrated in Fig. R6a. The Fermi level shift mechanism can be well supported by our DFT calculations of RuO₂ and Ru_{0.5}Cr_{0.5}O₂ as shown in Fig. R6b, and the experimental observations of the gradual decrease of the fractional valence of Cr-ions in Ru_{1-x}Cr_xO₂ from x = 0.1 to 0.3 (Fig. R6c).

(2) Our DFT calculation results distinctly indicate that the antiparallel coupling between the nearest neighboring ions in the Cr-doped RuO₂ lattice, as discussed in the response to Reviewer #1's Comment 1. Given that the doped Cr ions should be randomly located at the two sublattices, the collinear antiferromagnetic state with vanishingly small net moment would be stabilized as long as the doping level is not quite large ($\leq 20\%$).

(3) The Curie-Weiss fitting of the $\chi^{-1}-T$ curves give a Weiss temperature θ_w of -75 K for Ru_{0.8}Cr_{0.2}O₂ (Fig. R6d), and furthermore, the $M-H$ curve exhibits an almost two orders of magnitude smaller magnetization than ferromagnetic CrO₂ at 7 T. Both

observations emphasize the antiferromagnetic nature. Furthermore, we note that the ratio $f = \theta_W/T_N$ is frequently used as a rough measure of the degree of spin frustration; for instance, in such a collinear antiferromagnet that the magnetic interaction energy is optimized for any magnetic bond, f is usually about 2, while in a complicated noncollinear magnetic order that each magnetic bond cannot fully gain the magnetic interaction energy, f can be 10 or more [12]. In our case, the value of f is 1.5–1.8 (here T_N is 40–50 K, which is determined by the AHE), indicating that the nearest neighbor interaction energy $JS \cdot S$ is near its maximum for all magnetic bonds (i.e., unfrustrated). Thus, the low value of f is also consistent with the scenario of the collinear antiferromagnetic order.

Based on these discussions, we conclude that the $\text{Ru}_{0.8}\text{Cr}_{0.2}\text{O}_2$ maintains the collinear antiferromagnetic state. These discussions have been added to the revised manuscript.

Fig. R6. **a**, Illustration of the charge reconstruction and reduction of Cr from +4 to +3 valence state. **b**, XAS of the $\text{Ru}_{1-x}\text{Cr}_x\text{O}_2$ films around the Cr L-edges. The spectra of $\text{Cr}^{+3.25}$ and $\text{Cr}^{+3.5}$ are obtained from the reference with a $\text{La}_{0.75}\text{Sr}_{0.25}\text{CrO}_3$ and $\text{La}_{0.5}\text{Sr}_{0.5}\text{CrO}_3$, respectively. **c**, Calculated density of states (PDOS) of the RuO_2 and $\text{Ru}_{0.5}\text{Cr}_{0.5}\text{O}_2$ in the paramagnetic phase. The Ru-2 sites for both components possess identical PDOS with Ru-1. **d**, Temperature dependent magnetic susceptibility (χ) with Curie-Weiss fittings. Inset, the effective moments obtained from the fittings.

Comment 5: The authors claim that the $x=0.3$ case is likely ferrimagnetic. I don't

understand why that should necessarily be the case. In systems where the AHE is allowed by symmetry, the symmetry also allows existence of a net magnetic moment, since the symmetry requirements for AHE and a net moment are the same. Such a moment may be small or even zero and is not necessarily related to the AHE but is always allowed by symmetry. Thus, if the zero field AHE is observed it is not surprising that a net moment is also observed.

Response:

We thank the reviewer for this comment. In this work, we mainly focus on the AHE emerging in the collinear antiferromagnetic phase with vanishingly small net magnetization, i.e., the case of $x = 0.2$. We agree with the reviewer's symmetry argument that both cases with and without a net magnetization are allowed in the antiferromagnetic AHE. For example, the Mn_3Sn possesses a small canting moment at zero field while MnTe does not.^{[14],[15]} Thus, from the symmetry point of view, one may think that there is no need to make a particular distinction between $x = 0.2$ and $x = 0.3$. On the other hand, it appears to us that the case of $x = 0.3$ shows a distinct deviation from the antiferromagnetic state; for instance, the θ_w is positive, and the zero-field magnetization is greatly enhanced. These observations suggest that the magnetic state of $x = 0.3$ is more likely to be a ferrimagnetic state, in striking contrast to the non-collinear antiferromagnetic materials with small net moment. In fact, though we observed a larger σ_{xy}^{AHE} in the $x = 0.3$ material, it is more nontrivial to distinguish whether the σ_{xy}^{AHE} originates from the spin texture or the uniform moment (as in the case of ordinary ferromagnets). Thus, though we showed the antiferromagnetic to ferrimagnetic phase transition from $x = 0.2$ to 0.3 , we determined not to discuss the AHE for $x = 0.3$. We feel that the emergence of the zero-field AHE in the presence of an appreciable zero-field uniform moment, as in the case of $x = 0.3$, is less surprising compared with the case of $x = 0.2$, in which the zero-field AHE appears with the vanishingly small zero-field moment.

Comment 6: I'm not sure if the term "spontaneous AHE" will be clear to readers. To me it seems strange and misleading. I understand what the authors mean by it: the fact that the AHE shows hysteresis and is thus observed even at zero field. Yet, as far as I can see such term has not been commonly used in the previous literature. Rather the

term “spontaneous Hall effect” has been historically used as another name for the “anomalous Hall effect”. This is not used much anymore, though occasionally it is still used. Thus, I think this may confuse readers since “spontaneous Hall effect” and “anomalous Hall effect” have historically meant the same thing. Furthermore, the term “spontaneous AHE” might imply that the regular AHE is not spontaneous, which is certainly not the case in general. To me it’s also a strange expression because I would argue that even in previous experiments on RuO₂ the observed AHE is spontaneous. Although magnetic field is needed to observe it, this is only because the magnetic moments need to be tilted out of the c-axis. The origin of the effect still is a spontaneous symmetry breaking due to the magnetic order and the tilting could in principle be achieved by other means.

Response:

We thank the reviewer for this comment. We have modified the title to “Emergent zero-field anomalous Hall effect in a reconstructed rutile antiferromagnetic metal”, to highlight the scientific merit compared to the previous report on the high-field AHE in pure RuO₂. Furthermore, to avoid confusions, we have removed “spontaneous AHE” and revised the manuscript accordingly.

Comment 7: There are quite significant errors throughout the manuscript, mainly in the introduction, regarding the symmetry. The authors claim that the existence of AHE is determined by whether the material has a time-reversal symmetry. That is not correct. It is true that the AHE cannot exist in materials with time-reversal symmetry, however, time-reversal symmetry is broken by any magnetic order including antiferromagnetic orders even those that do not allow the AHE. It is not true that antiferromagnets invariant under time-reversal followed by lattice translation have time-reversal symmetry. Rather such systems have a symmetry operation that is composed of time-reversal plus translation. Such symmetry operation is distinct from the time-reversal, but like the time-reversal it prohibits the existence of the AHE. There can be other symmetry operations that prohibit the AHE, in particular the time-reversal plus spatial inversion. Even symmetry operations that have nothing to do with time-reversal can prohibit AHE, for example, no AHE can exist in the presence of two rotation axes.

In RuO₂ the time-reversal plus translation and the time-reversal plus inversion

symmetries are broken, but when the magnetic moments are along c axis, the AHE is still prohibited by symmetry, due to existence of other symmetry operations. When the magnetic moments are moved away from the c-axis then the AHE is allowed by symmetry.

Response:

We appreciate the insightful comments from the referee. In the previous version, we focused on whether the magnetic point group (in which the translation operations are set as zero, in contrast to the magnetic space group) has time-reversal symmetry (TRS), because the AHE is forbidden when TRS is present in the magnetic point group. We agree that the absence of TRS in the magnetic point group is merely a necessary condition for a finite AHE, and that an AHE may still be not allowed if other symmetry operations, such as two rotation axis (C_{2x} and C_{2y}) and (\mathcal{C}_{4z}) are present. This is the case of $P4_2'/mnm'$ (RuO_2 with the Néel vector along the [001] axis), as we discussed in our response to Comment 1.

In the revised version, care was taken not to misuse the term "time-reversal symmetry breaking", and the difference between magnetic point group and magnetic space group for the symmetry analysis has been further clarified to avoid confusions. We also specified that RuO_2 with the Néel vector along the [001] axis is a system in which TRS is absent in the magnetic point group but the AHE is prohibited by the other rotation symmetry operations.

Comment 8: The authors explain the existence of AHE and its symmetry properties using the Berry curvature. This is commonly done, but this should only be seen as illustration of how AHE can originate. There are other mechanisms beyond the Berry curvature and the existence of AHE is determined by symmetry arguments that are independent of the Berry curvature.

Furthermore, the authors claim on lines 206 and 227, that the origin of the observed AHE is the Berry curvature. As far as I can see, there is no basis for such a claim, and it is entirely possible that the observed AHE has an extrinsic origin in their case.

Response:

We thank the reviewer for this comment. We agree that apart from the intrinsic

Berry curvature, the extrinsic side-jump and skew-scattering can also be the microscopic mechanisms to generate AHE in the magnetic materials [2],[3]. As we discussed in the response to the Comment-3 of Reviewer #1, the skew scattering is mostly relevant for materials with very high longitudinal conductivity ($\sigma_{xx0} > 10^6$ S/cm), and this mechanism can thus be ruled out in our materials ($10^3 < \sigma_{xx0} < 10^4$ S/cm). Besides, we have tried to distinguish the side-jump and intrinsic terms of the canting moment-induced anomalous Hall conductivity through the scaling of $\sigma_{xy}^M \sim \sigma_{xx}(T)^2$ within the series of films (in the intermediate regime). We thus found that σ_{xy}^M due to the intrinsic Berry curvature is 14 S/cm, whereas the σ_{xy}^M due to the extrinsic side-jump mechanism is 10 S/cm.

On the other hand, we didn't find a model to quantitatively distinguish the side-jump and intrinsic Berry curvature contributions to the collinear-antiferromagnetic-order-induced σ_{xy}^{AF} . However, we speculate that the relative magnitudes of the intrinsic Berry curvature and extrinsic side-jump contributions to σ_{xy}^{AF} are similar to the case of σ_{xy}^M .

In the revised manuscript, we have added the symmetry discussions in the introduction part to avoid the confusion of considering Berry curvature as the only microscopic mechanism to generate the anomalous Hall voltage. We have also added the discussions of the microscopic mechanisms to the reversed manuscript.

Comment 9: The authors claim that the distinguishing feature of their experiments compared to the previous experiments where hysteretic AHE in antiferromagnets was observed, is that the previous experiments used antiferromagnets with geometrical frustration. My understanding is that at least in the case of Mn₃Sn or similar antiferromagnets there is in fact no geometrical frustration. Frustration would exist in these systems if the magnetic moments were collinear, but the non-collinear order that is observed is not frustrated. Perhaps more importantly, even if there was a geometrical frustration, the manuscript suggests that this frustration is somehow crucial for the existence of the AHE, which is not the case. I also find the claim that the RuO₂ is a “much simpler” system than the previously studied antiferromagnets dubious. Simplicity can of course mean different things, but the Mn₃Sn antiferromagnet, for example, also has quite a simple magnetic and crystal structure.

I also point out that recently there has been observation of AHE in MnTe [PRL 130, 036702 (2023)], which is a collinear antiferromagnet similar to RuO₂ that certainly

has no frustration. This work needs to be cited.

Response:

We thank the reviewer for these comments. In the previous manuscript, we compared the collinear and noncollinear antiferromagnetic materials in the introduction. We respectfully point out that the kagome and pyrochlore lattice structures, such as Mn_3Sn and $\text{Nd}_2\text{Mo}_2\text{O}_7$, are very frequently referred to as a frustrated lattice geometry [12], [16], [17], [18],[19]. In our manuscript, we used the “frustration” to mean that a collinear antiferromagnetic order cannot be the ground state because of the competition of several magnetic interactions. We do not mean that the frustration prohibits a noncollinear magnetic order; for instance, in the literature,[20] the authors say that a helical magnetic order and skyrmion-lattice order are realized via the frustrated symmetric exchange interaction. What we wanted to mention is that the AHEs in antiferromagnets of previous studies were found mainly in noncollinear magnetic orders that emerged in so-called frustrated lattices. Of course, we agree that the frustrated lattice geometry is not crucial for the emergence of the AHE, as demonstrated in the (Cr-doped) RuO_2 system. Magnetic point groups in which AHE is allowed can also be realized in non-frustrated lattices. We have changed the related description to avoid any confusions.

We agree that the “simpler” is too subjective and not a good description as comparing the collinear and noncollinear antiferromagnets, and we have also revised the manuscript to avoid giving the impression that magnetic frustration is crucial for AHE.

We thank the reviewer’s reminder of the newly published paper about the AHE in antiferromagnetic semiconductor MnTe , which exhibits a small σ_{xy}^{AHE} of 0.1 S/cm without magnetization[15]. This paper was published after our submission in Dec. 2022, and thus was not cited. Now, we have cited this paper in the revised manuscript.

Comment 10: Is there a reason why the experiments did not include the case of pure RuO_2 ? It would be interesting to have this for comparison.

Response:

As for the magnetization measurements, we note the $\chi^{-1}-T$ fitting should use the data above the magnetic ordering transition temperature, while the RuO_2 is reported to

have a T_N much higher than the room temperature, [13] which is beyond the limit of our equipment.

As for the Hall effect, in contrast to the doped cases, the pure RuO_2 shows a merely linear dependence to the magnetic field within ± 9 T in our measurements (Fig. R7). In such a low-field regime, both the ordinary Hall effect (OHE) and the AHE (in the pure RuO_2 , due to the field-induced tilting of the Néel vector off the [001] direction) are expected to behave linearly. Therefore, it is impossible to distinguish the OHE and AHE. To distinguish them, a high magnetic field of ~ 50 T is needed, and the result has been already reported by Z. Feng et al. [7]

Fig. R7. Magnetic field dependent Hall resistances (R_{yx}) of the RuO_2 films grown on TiO_2 with c axis along [100], [110], and [101] at 3 K.

References :

-
- [1] Zhang, K. H. L. et al. Hole-induced insulator-to-metal transition in $\text{La}_{1-x}\text{Sr}_x\text{CrO}_3$ epitaxial films. *Phys. Rev. B* **91**, 155129 (2015).
- [2] Miyasato, T. et al. Crossover behavior of the anomalous Hall effect and anomalous Nernst effect in itinerant ferromagnets. *Phys. Rev. Lett.* **99**, 086602 (2007).
- [3] Nagaosa, N., Sinova, J., Onoda, S., MacDonald, A. H. & Ong, N. P. Anomalous Hall effect. *Rev. Mod. Phys.* **82**, 1539-1592 (2010).
- [4] Fujishiro, Y. et al. Giant anomalous Hall effect from spin-chirality scattering in a chiral magnet. *Nat. Commun.* **12**, 317 (2021).
- [5] Tian, Y., Ye, L., & Jin, X. Proper scaling of the anomalous Hall effect. *Phys. Rev. Lett.* **103**, 87206 (2009).
- [6] Li, Y. F. et al. Robust formation of skyrmions and topological Hall effect anomaly in epitaxial thin films of MnSi . *Phys. Rev. Lett.* **110**, 117202 (2013).
- [7] Feng, Z. et al. An anomalous Hall effect in altermagnetic ruthenium dioxide. *Nat. Electron.* **5**, 735-743 (2022).

-
- [⁸] Šmejkal, L., González-Hernández, R., Jungwirth, T. & Sinova, J. Crystal time-reversal symmetry breaking and spontaneous Hall effect in collinear antiferromagnets. *Sci. Adv.* **6**, eaaz8809 (2020).
- [⁹] Šmejkal, L. et al. Anomalous Hall antiferromagnets. *Nat. Rev. Mater.* **7**, 482-496 (2022).
- [¹⁰] Blundell, S. Magnetism in Condensed Matter (Oxford University Press New York, 2001).
- [¹¹] Mugiraneza, S. & Hallas, A. M. Tutorial: A beginner's guide to interpreting magnetic susceptibility data with the Curie-Weiss law. *Commun. Phys.* **5**, 1-12 (2022).
- [¹²] Greedan, J. E. Geometrically frustrated magnetic materials. *J. Mater. Chem.* **11**, 37 (2001).
- [¹³] Berlijn, T. et al. Itinerant antiferromagnetism in RuO₂. *Phys. Rev. Lett.* **118**, 077201 (2017).
- [¹⁴] Nakatsui, S., Kiyohara, N. & Higo, T. Large anomalous Hall effect in a non-collinear antiferromagnet at room temperature. *Nature* **527**, 212-215 (2015).
- [¹⁵] Gonzalez Betancourt, R. D. et al. Spontaneous anomalous Hall effect arising from an unconventional compensated magnetic phase in a semiconductor. *Phys. Rev. Lett.* **130**, 036702 (2023).
- [¹⁶] Takeuchi, Y. et al. Chiral-spin rotation of non-collinear antiferromagnet by spin-orbit torque. *Nat. Mater.* **20**, 1364-1370 (2021).
- [¹⁷] Taguchi, Y. et al. Spin chirality, berry phase, and anomalous hall effect in a frustrated ferromagnet. *Science* **291**, 2573-2576 (2001).
- [¹⁸] Kim, W. J. et al. Strain engineering of the magnetic multipole moments and anomalous Hall effect in pyrochlore iridate thin films. *Sci. Adv.* **6**, eabb1539 (2020).
- [¹⁹] Chen, H., Niu, Q. & MacDonald, A. H. Anomalous Hall effect arising from noncollinear antiferromagnetism. *Phys. Rev. Lett.* **112**, 017205 (2014).
- [²⁰] Okubo, T. et al. Multiple-*q* states and the skyrmion lattice of the triangular-lattice Heisenberg antiferromagnet under magnetic fields. *Phys. Rev. Lett.* **108**, 017206 (2012).

Reviewers' Comments:

Reviewer #1:

Remarks to the Author:

In the revised manuscript and response letter, Meng Wang et al. have comprehensively answered all the comments raised by me. I particularly appreciate the reviewer for adding more reliable DFT calculations and analyses on the origin of the anomalous Hall effect. The quality of the manuscript is improved considerably. Therefore, I recommend this paper be published in Nature Communications as it is.

Reviewer #2:

Remarks to the Author:

The manuscript has been improved considerably and the interpretation now seems reasonable. However, some issues remain.

Although the interpretation of the experiments seems much more reasonable than before, I'm still not convinced that such strong conclusions can be made. The central result of this work is the detection of anomalous Hall effect (AHE) in collinear antiferromagnet in absence of magnetic field due to a change of the easy axis caused by doping. In my opinion the results clearly show that the material is antiferromagnetic, but the rest is less clear.

I don't think there is any evidence that the material is in fact collinear. The arguments given for this are quite weak. DFT cannot be seen as a strong evidence of the magnetic order since it can give wrong results. Ultimately, experiment is always necessary. Furthermore, the arguments authors give based on DFT do not really show that the lowest energy state is the collinear state, but at best suggest it. I'm also very skeptical about the reasoning based on the Curie-Weiss fitting. I've found a measurements of the Curie-Weiss temperature of a non-collinear antiferromagnet on a frustrated kagome lattice that show the ratio between the Curie-Weiss temperature and the Néel temperature less than 1: <https://arxiv.org/pdf/1904.05678.pdf>. I really doubt that this is some strong rule that can be used to determine whether magnetic order is collinear. I also don't think that the non-collinearity only originates from frustration.

I don't think the magnetic order in RuO₂ is completely clear either, see for example discussion in <https://arxiv.org/pdf/2306.12130.pdf>. The situation in the Cr doped RuO₂ is even worse since no direct determination of the magnetic structure is available and the doping is large so the magnetic structure could easily change from RuO₂. The large change of the Néel temperature suggests a large change of the magnetic interactions, which of course doesn't mean that the magnetic order is different, but it shows that the doping changes the material properties significantly.

This means that also it's not clear that the reason for the existence of AHE is a change of the easy axis. The result suggest that the Hall vector lies along the [110] direction, but that doesn't necessarily mean that the Néel vector lies along the [110]. For example in MnTe, the magnetic moments lie in the (001) plane, but the Hall vector is along the [001] direction.

Furthermore, without knowing the magnetic order and how it reacts to the magnetic field, it is not even clear that the hysteretic signal is necessarily due to AHE. This is because AHE is by definition effect odd under time-reversal and thus to show that the effect is AHE it is necessary to show that it is an effect odd under reversal of all magnetic moments (and magnetic field). However, without knowing how the magnetic order reacts to the magnetic field it is not possible to say that the two states at H=0 correspond to states with opposite magnetic moments. This is crucial since if this is not the case, the signal could be simply due to the anisotropic magnetoresistance. Certainly in a non-collinear system, but in principle also in collinear, it could definitely happen that the magnetic field switches between two states that are not related by reversal of all moments.

This is in general a complicated problem, which is present also in many other (published) works from the field. It is probably not realistic for these problems to be solved within this work. Overall the interpretation seems reasonable to me and I believe the work is interesting and will be useful

to other researchers. Thus in my opinion, this uncertainty should not necessarily prevent the work from being published, however, I would suggest making the uncertainty more clear in the manuscript. The interpretation of the experiments is based on several assumptions: that the magnetic order is collinear, that the magnetic field reverses all the magnetic moments and that the doping is changing the easy axis to [110]. There are some justifications of these assumptions, but no direct evidence and this should be clearly explained in the manuscript.

Below, I give some more specific comments:

1. The authors now cite the AHE experiments in MnTe, however, not in the introduction. This is misleading since in the introduction the authors claim that the AHE has been measured in non-collinear antiferromagnets and then discuss RuO₂ suggesting that there is no collinear system where AHE in absence of magnetic field has been measured.

2. In the introduction, the authors define the Hall vector. Here it should be explained that this is only taking into account the anti-symmetric part (or equivalently the part that is odd under time-reversal) of the conductivity tensor and not the full conductivity tensor.

3. It may be true that the magnetic moments are increasing because of doping, however, the evidence is rather weak. The authors compare results from Curie-Weiss fitting for the doped material with results from neutron scattering for pure RuO₂. Furthermore, the small moment for pure RuO₂ is questionable. I note that although the authors claim that the moment is very tiny in pure RuO₂, as far as I can understand it, their own calculations (as well as calculations by other people studying this material) use magnetic moments two orders of magnitude larger. Thus although the doping might very well increase the magnetic moments, the evidence of it is rather weak and even if it does happen it is quite likely that the increase is much smaller than the manuscript suggests. The doping decreases the Néel temperature significantly so it's certainly not making the magnetic order more robust.

4. More importantly, the authors claim that the "impact of collinear antiferromagnetic order on the transport properties is more observable due to the enhancement of the local magnetic moments", however, they measure only a very tiny AHE of few S/cm. In contrast the magnitude of AHE found for pure RuO₂ in calculations and experiment is two orders of magnitude larger (Ref. 17). This argument is thus very questionable. The authors should either justify it or remove it.

5. The authors argue that the absence of Kerr effect is an evidence of the antiferromagnet order. However, the symmetry conditions of the Kerr effect are the same as that of AHE and thus if AHE is allowed the Kerr effect must also be allowed by symmetry. Thus the absence of Kerr cannot be seen as evidence of the magnetic order being antiferromagnetic.

I can recommend the paper to be published if these issues are resolved.

We appreciate the constructive comments and suggestions from the reviewers. Accordingly, we have further improved our manuscript as highlighted in purple color. In the following, we respond to the comments of the reviewers, point by point.

Reviewer #1 (Remarks to the Author):

In the revised manuscript and response letter, Meng Wang et al. have comprehensively answered all the comments raised by me. I particularly appreciate the reviewer for adding more reliable DFT calculations and analyses on the origin of the anomalous Hall effect. The quality of the manuscript is improved considerably. Therefore, I recommend this paper to be published in Nature Communications as it is.

Response:

We thank the reviewer for recommending our revised manuscript to be published in Nature Communications.

Reviewer #2 (Remarks to the Author):

The manuscript has been improved considerably and the interpretation now seems reasonable. However, some issues remain.

Although the interpretation of the experiments seems much more reasonable than before, I'm still not convinced that such strong conclusions can be made. The central result of this work is the detection of anomalous Hall effect (AHE) in collinear antiferromagnet in absence of magnetic field due to a change of the easy axis caused by doping. In my opinion the results clearly show that the material is antiferromagnetic, but the rest is less clear.

I don't think there is any evidence that the material is in fact collinear. The arguments given for this are quite weak. DFT cannot be seen as a strong evidence of the magnetic order since it can give wrong results. Ultimately, experiment is always necessary. Furthermore, the arguments authors give based on DFT do not really show that the lowest energy state is the collinear state, but at best suggest it. I'm also very skeptical about the reasoning based on the Curie-Weiss fitting. I've found a measurements of the Curie-Weiss temperature of a non-collinear antiferromagnet on a frustrated kagome lattice that show the ratio between the Curie-Weiss temperature and

the Néel temperature less than 1: <https://arxiv.org/pdf/1904.05678.pdf>. I really doubt that this is some strong rule that can be used to determine whether magnetic order is collinear. I also don't think that the non-collinearity only originates from frustration.

I don't think the magnetic order in RuO₂ is completely clear either, see for example discussion in <https://arxiv.org/pdf/2306.12130.pdf>. The situation in the Cr doped RuO₂ is even worse since no direct determination of the magnetic structure is available and the doping is large so the magnetic structure could easily change from RuO₂. The large change of the Néel temperature suggests a large change of the magnetic interactions, which of course doesn't mean that the magnetic order is different, but it shows that the doping changes the material properties significantly.

This means that also it's not clear that the reason for the existence of AHE is a change of the easy axis. The result suggest that the Hall vector lies along the [110] direction, but that doesn't necessarily mean that the Néel vector lies along the [110]. For example in MnTe, the magnetic moments lie in the (001) plane, but the Hall vector is along the [001] direction.

Furthermore, without knowing the magnetic order and how it reacts to the magnetic field, it is not even clear that the hysteretic signal is necessarily due to AHE. This is because AHE is by definition effect odd under time-reversal and thus to show that the effect is AHE it is necessary to show that it is an effect odd under reversal of all magnetic moments (and magnetic field). However, without knowing how the magnetic order reacts to the magnetic field it is not possible to say that the two states at H=0 correspond to states with opposite magnetic moments. This is crucial since if this is not the case, the signal could be simply due to the anisotropic magnetoresistance. Certainly in a non-collinear system, but in principle also in collinear, it could definitely happen that the magnetic field switches between two states that are not related by reversal of all moments.

This is in general a complicated problem, which is present also in many other (published) works from the field. It is probably not realistic for these problems to be solved within this work. **Overall the interpretation seems reasonable to me and I believe the work is interesting and will be useful to other researchers. Thus in my opinion, this uncertainty should not necessarily prevent the work from being published, however, I would suggest making the uncertainty more clear in the manuscript.** The interpretation of the experiments is based on several assumptions:

that the magnetic order is collinear, that the magnetic field reverses all the magnetic moments and that the doping is changing the easy axis to [110]. There are some justifications of these assumptions, but no direct evidence and this should be clearly explained in the manuscript.

Response:

We sincerely thank the reviewer for carefully examining our revised manuscript and believing that our work is interesting and useful for researchers to be published.

We agree with the reviewer's comment that although the antiferromagnetic (AFM) state in $\text{Ru}_{0.8}\text{Cr}_{0.2}\text{O}_2$ has been supported by our observations, its details, such as collinearity (and thus the Néel vector), have not been fully revealed. Following the reviewer's suggestions, we have noted this weakness explicitly and discussed the implications of our observations, being careful not to rule out possibilities other than the collinear AFM order.

Here, we'd like to respond to some points of the above comments:

(1) We thank the reviewer's comment on the relationship between the Curie-Weiss temperature and the magnetic frustration. The arxiv paper discusses $\text{Mn}_3(\text{Ni}_{1-x}\text{Cu}_x)\text{N}$ antiperovskite (<https://arxiv.org/pdf/1904.05678.pdf>), which however could host several magnetic transitions, including collinear antiferromagnetic orders near the T_N and non-collinear orders at low temperatures; furthermore, the magnetism is very sensitive to the strain in films [1]. Given that the paper does not show evidence of non-collinear order in their films, we are not sure whether this paper can be considered a counterexample demonstrating a non-collinear antiferromagnet with a small $|\theta_W/T_N|$. Nevertheless, we agree that the Curie-Weiss analysis is not sufficiently strong to conclude the collinearity of the antiferromagnetism. We have noted this point in the revised manuscript.

(2) The antiferromagnetic nature in RuO_2 seems becoming controversial, as discussed in <https://arxiv.org/pdf/2306.12130.pdf>. Nevertheless, there is no doubt that the rutile structure is quite interesting in that it can exhibit the AHE in the presence of a collinear AFM order. Although the possibility of the collinear AFM state in our $\text{Ru}_{0.8}\text{Cr}_{0.2}\text{O}_2$ films is indicated by our observations of the anisotropic magnetization, zero-field AHE and crystalline-dependent AHC, we agree that these observations are not strong enough to rule out other possibilities. Nevertheless, considering that AFM

order may be realized in other Cr-doped rutile compounds, such as $\text{Cr}_x\text{Re}_{1-x}\text{O}_2$,^[2] we believe that our study will stimulate further exploration of the AHE in rutile AFM materials.

(3) We agree with the reviewer that although our magnetometry results (Fig. 3) indicate that the Néel vector is likely along the [110] direction rather than [001], some other directions cannot be completely ruled out. On the other hand, the difference in the transport behaviors by a factor of $\sin 45^\circ$ between films grown along [100] and [110] can be well explained by assuming the Hall vector along the [110] direction. Moreover, the Hall vector along the [110] direction is reasonably expected if the collinear AFM order with the Néel vector along the [110] direction is realized. In the revised manuscript, we have noted that our observations can be consistently understood if the collinear AFM order with the Néel vector along the [110] direction is assumed, being careful not to rule out other possibilities.

(4) We agree that although we clearly observed the two states at $H = 0$ (positive AHC and negative AHC), it is less clear that these two states are really related with the reversal of all moments. In the revised manuscript, we noted that the observation of the two states at $H = 0$ can be understood if the moment reversal of the collinear AFM order is assumed, but we were careful not to rule out other possibilities.

Following the reviewer's suggestion, in the last paragraph of the introduction section, we have explicitly noted that the present experimental results are interpreted within the assumptions mentioned above.

(Below is the copy from the main text)

It should be noted that the collinear antiferromagnetism in RuO₂ has been questioned quite recently.²⁷ Nevertheless, considering that more previous experiments support the collinear antiferromagnetism^{17,19,24-26}, we design the experiment by postulating that the magnetism of RuO₂ at the ground state is the collinear antiferromagnetism with the Néel vector along [001] and interpret the experimental results within an assumption that a small amount of Cr-doping does not change the collinear antiferromagnetism but can modulate the direction of the Néel vector. Within this approach, our magnetometry suggests that the direction of the Néel vector in the Ru_{0.8}Cr_{0.2}O₂ film is changed into [110]. Concomitantly, we find that the Ru_{0.8}Cr_{0.2}O₂ film exhibits an appreciable zero-field AHE with hysteretic behaviour while the net magnetization is vanishingly small. These observations are well explained by considering that the collinear antiferromagnetism with the Néel vector along [110] is realized and that the magnetic field switches the two collinear antiferromagnetic states that are related by time-reversal operation.↵

↵

Results↵

DFT calculations on the impact of Cr-doping↵

Below are our point-to-point responses to the reviewer's comments.

Below, I give some more specific comments:

1. The authors now cite the AHE experiments in MnTe, however, not in the introduction. This is misleading since in the introduction the authors claim that the AHE has been measured in non-collinear antiferromagnets and then discuss RuO₂ suggesting that there is no collinear system where AHE in absence of magnetic field has been measured.

Response:

We thank the reviewer for this comment. Following the suggestion, we have discussed the AHE of MnTe and other collinear antiferromagnetic materials in the introduction.

We'd like to point out that the rutile oxide has attracted attentions as a model system of AHE in the presence of a collinear antiferromagnetic (or altermagnetic) order,^[3] and we believe that our work demonstrates a new approach to manipulating its AHE.

2. In the introduction, the authors define the Hall vector. Here it should be explained that this is only taking into account the anti-symmetric part (or equivalently the part that is odd under time-reversal) of the conductivity tensor and not the full conductivity tensor.

Response:

According to the reviewer's comment, we have defined Hall vector more carefully.

3. It may be true that the magnetic moments are increasing because of doping, however, the evidence is rather weak. The authors compare results from Curie-Weiss fitting for the doped material with results from neutron scattering for pure RuO₂. Furthermore, the small moment for pure RuO₂ is questionable. I note that although the authors claim that the moment is very tiny in pure RuO₂, as far as I can understand it, their own calculations (as well as calculations by other people studying this material) use magnetic moments two orders of magnitude larger. Thus although the doping might very well increase the magnetic moments, the evidence of it is rather weak and even if it does happen it is quite likely that the increase is much smaller than the manuscript suggests. The doping decreases the Néel temperature significantly so it's certainly not making the magnetic order more robust.

Response:

We thank the reviewer for this comment. We note that the antiferromagnetic local moment in RuO₂ is estimated by neutron diffraction, while our results are derived from the susceptibility measurements. We also note that although the prior neutron diffraction in bulk RuO₂ reported a high Néel temperature ($T_N > 300$ K),^[4] the T_N values vary from paper to paper; for instance, Feng et al. reported an observable high-field AHE only below 150 K in the RuO₂ films^[5]. Thus, the real T_N of RuO₂ seems controversial and may not be so high. Considering that the previous neutron-diffraction results (quite small local moment $0.05 \mu_B$ with high T_N) is becoming questionable, we have changed our mind that our results should not be compared too much with the neutron results in RuO₂. Therefore, in the revised manuscript, we have removed the direct comparison of the local moment between RuO₂ and Ru_{1-x}Cr_xO₂.

Besides, our experimental and calculated results indicate the possibility of the collinear antiferromagnetic order in $\text{Ru}_{0.8}\text{Cr}_{0.2}\text{O}_2$, but stronger evidence is still needed. In the revised manuscript, we were thus careful to point out that our observations do not completely rule out any possibility other than the collinear antiferromagnet.

4. More importantly, the authors claim that the “impact of collinear antiferromagnetic order on the transport properties is more observable due to the enhancement of the local magnetic moments”, however, they measure only a very tiny AHE of few S/cm. In contrast the magnitude of AHE found for pure RuO_2 in calculations and experiment is two orders of magnitude larger (Ref. 17). This argument is thus very questionable. The authors should either justify it or remove it.

Response:

As we discussed in the previous version, in addition to enhancing local moments, Cr-doping also enhances scattering and, more importantly, changes the Fermi level. The value of AHC should be a result of all three factors, and thus, we thought that it was a bit misleading to emphasize only the magnitude of local magnetization. Following the reviewer’s suggestion, we have removed such descriptions in the revised manuscript.

5. The authors argue that the absence of Kerr effect is an evidence of the antiferromagnet order. However, the symmetry conditions of the Kerr effect are the same as that of AHE and thus if AHE is allowed the Kerr effect must also be allowed by symmetry. Thus the absence of Kerr cannot be seen as evidence of the magnetic order being antiferromagnetic.

Response:

We agree with the reviewers’ comment that the Kerr effect in our material should be allowed from a symmetry point of view because we have observed finite AHE at $H = 0$. However, it is known that the Kerr rotation angle in a magnetic material could be sensitive to the incident wavelength, and it could be too small to be detected at a selected wavelength ^[6]. Our motivation of the Kerr-rotation microscopy was to observe the positive-AHC and negative-AHC domains, but we could not observe them at our wavelength, and we could not change the wavelength of our Kerr equipment.

Unfortunately, our Kerr effect measurements did not provide any useful implications, so we removed the description of the Kerr measurements.

I can recommend the paper to be published if these issues are resolved.

Response:

We thank the reviewer for these comments, which have helped us to further improve our manuscript. We hope that our revised manuscript will inspire further studies on rutile materials or their superlattices to design and manipulate the antiferromagnetism and resulting AHE.

References :

- [¹] Boldrin, D. et al. The biaxial strain dependence of magnetic order in spin frustrated Mn_3NiN thin films. *Adv. Funct. Mater.* **29**, 1902502 (2019).
- [²] Mikhailova, D. et al. $Cr_xRe_{1-x}O_2$ oxides with different rutile-like structures: changes in the electronic configuration and resulting physical properties. *J. Solid State Chem.* **182**, 1506 (2009).
- [³] Šmejkal, L., Sinova, J. & Jungwirth, T. Emerging research landscape of altermagnetism. *Phys. Rev. X* **12**, 040501 (2022).
- [⁴] Berlijn, T. et al. Itinerant antiferromagnetism in RuO_2 . *Phys. Rev. Lett.* **118**, 077201 (2017).
- [⁵] Feng, Z. et al. An anomalous Hall effect in altermagnetic ruthenium dioxide. *Nat. Electron.* **5**, 735-743 (2022).
- [⁶] Feng, W., Guo, G.-Y., Zhou, J., Yao, Y. & Niu, Q. Large magneto-optical Kerr effect in noncollinear antiferromagnets Mn_3X ($X = Rh, Ir, Pt$). *Phys. Rev. B* **92**, 144426 (2015).

Reviewers' Comments:

Reviewer #2:

Remarks to the Author:

The authors have replied to all of my comments and I can thus recommend the manuscript to be published.

Reviewer #2 (Remarks to the Author):

The authors have replied to all of my comments and I can thus recommend the manuscript to be published..

Response:

We thank the reviewer for recommending our revised manuscript to be published in Nature Communications.